# Interaction of amisulpride with GLUT1 at the blood-brain barrier. Relevance to Alzheimer's disease

Sevda T. Boyanova[1], Ethlyn Lloyd-Morris[1], Christopher Corpe[2], Khondaker Miraz Rahman[1], Doaa B. Farag[3], Lee K. Page[1], Hao Wang[1], Alice L. Fleckney[1], Ariana Gatt[4], Claire Troakes[5], Gema Vizcay-Barrena[6], Roland Fleck[6], Suzanne J. Reeves[7], Sarah A. Thomas[8]*

1 King's College London, Institute of Pharmaceutical Science, London, United Kingdom, 2 King's College London, Department of Nutritional Sciences, School of Life Course Sciences, Faculty of Life Sciences and Medicine, London, United Kingdom, 3 Faculty of Pharmacy, Misr International University, Cairo, Egypt, 4 King's College London, Wolfson Centre for Age Related Disease, London, United Kingdom, 5 King's College London, London Neurodegenerative Diseases Brain Bank, IoPPN, London, United Kingdom, 6 King's College London, Centre for Ultrastructural Imaging, London, United Kingdom, 7 Faculty of Brain Sciences, University College London, London, United Kingdom, 8 King's College London, Department of Physiology, London, United Kingdom

* sarah.thomas@kcl.ac.uk

**Data Availability Statement:** All relevant data are within the paper and its Supporting Information files.

## Abstract

Blood-brain barrier (BBB) dysfunction may be involved in the increased sensitivity of Alzheimer's disease (AD) patients to antipsychotics, including amisulpride. Studies indicate that antipsychotics interact with facilitated glucose transporters (GLUT), including GLUT1, and that GLUT1 BBB expression decreases in AD. We tested the hypotheses that amisulpride (charge: +1) interacts with GLUT1, and that BBB transport of amisulpride is compromised in AD. GLUT1 substrates, GLUT1 inhibitors and GLUT-interacting antipsychotics were identified by literature review and their physicochemical characteristics summarised. Interactions between amisulpride and GLUT1 were studied using *in silico* approaches and the human cerebral endothelial cell line, hCMEC/D3. Brain distribution of [3H]amisulpride was determined using *in situ* perfusion in wild type (WT) and 5xFamilial AD (5xFAD) mice. With transmission electron microscopy (TEM) we investigated brain capillary degeneration in WT mice, 5xFAD mice and human samples. Western blots determined BBB transporter expression in mouse and human. Literature review revealed that, although D-glucose has no charge, charged molecules can interact with GLUT1. GLUT1 substrates are smaller (184.95 ±6.45g/mol) than inhibitors (325.50±14.40g/mol) and GLUT-interacting antipsychotics (369.38±16.04). Molecular docking showed beta-D-glucose (free energy binding: -15.39kcal/mol) and amisulpride (-29.04kcal/mol) interact with GLUT1. Amisulpride did not affect [14C]D-glucose hCMEC/D3 accumulation. [3H]amisulpride uptake into the brain (except supernatant) of 5xFAD mice compared to WT remained unchanged. TEM revealed brain capillary degeneration in human AD. There was no difference in GLUT1 or P-glycoprotein BBB expression between WT and 5xFAD mice. In contrast, caudate P-glycoprotein, but not GLUT1, expression was decreased in human AD capillaries versus controls. This study provides new details about the BBB transport of amisulpride, evidence that amisulpride

**Funding:** STB was a Guy's and St Thomas' Charity funded MRes-Ph.D. student. ELM was supported by the UK Medical Research Council and King's College London as a member of the MRC Doctoral Training Partnership in Biomedical Sciences ([www.ukri.org/councils/mrc/] MR/N013700/1). ALF was a BBSRC-CASE funded PhD student ([www.ukri.org/councils/bbsrc/] BB/L01534X/1). The study was supported, in part, by the King's College London Alzheimer's Research UK [www.alzheimersresearchuk.org/] network pump-priming grants (awarded to SAT, STB, CT, GVB, RF and SR) and a multi-user equipment grant (080268) from the Wellcome Trust [https://wellcome.org] awarded to SAT, Professors Francis, McMahon, Malcangio, and Rattray (all at King's College London). Human tissue samples were supplied by the Manchester Brain Bank and the London Neurodegenerative Diseases Brain Bank, which are both part of the Brains for Dementia Research programme, jointly funded by Alzheimer's Research UK and Alzheimer's Society [https://bdr.alzheimersresearchuk.org/] (Applicant SAT). This research was funded in whole, or in part, by the Wellcome Trust [080268]. For the purpose of Open Access, the author has applied a CC BY public copyright licence to any Author Accepted Manuscript version arising from this submission. The funders had no role in study design, data collection and analysis, decision to publish, or preparation of the manuscript.

**Competing interests:** The authors have declared that no competing interests exist.

**Abbreviations:** 3xTgAD, triple transgenic; Aβ, amyloid beta; AD, Alzheimer's disease; 5xFAD, five times familial Alzheimer's disease mouse model; ABC, Adenosine triphosphate binding cassette; AD, Alzheimer's disease; APP, amyloid precursor protein; ASN, asparagine; BBB, blood–brain barrier; BCA, bicinchoninic acid; BDR, Brain for Dementia Research; Caco2, immortalized cell line of human colorectal adenocarcinoma cells; Calu3, cultured human airway epithelial cells; CAS, chemical abstracts service number; $C_{max}$, maximum concentration after the drug has been administered; dpm, disintegrations per minute; FBS, foetal bovine serum; GAPDH, glyceraldehyde 3-phosphate dehydrogenase; GLN, glutamine; GLU, glutamate; GLUT1, glucose transporter 1; GLY, glycine; hCMEC/D3, immortalized human cerebral microvessel endothelial cell line; EK cells, human embryonic kidney cells; HL-60, Human promyelocytic leukaemia cell line; $IC_{50}$, half maximal inhibitory concentration; MATEs, multi-drug and toxin extrusion proteins; MW, molecular weight; OCT, organic cation transporter; OCTN,

interacts with GLUT1 and that BBB transporter expression is altered in AD. This suggests that antipsychotics could potentially exacerbate the cerebral hypometabolism in AD. Further research into the mechanism of amisulpride transport by GLUT1 is important for improving antipsychotics safety.

## Introduction

Psychosis, which most commonly presents as delusions, is highly prevalent (~50%) in people with Alzheimer's disease (AD) [1, 2] and is associated with poorer quality of life, a more rapid speed of cognitive and functional decline [2] and greater risk of institutionalisation [3]. Safe and effective prescribing of antipsychotic drugs is challenging in older people, particularly those with AD, due to their heightened susceptibility to the side effects of these drugs [4], including sedation, postural hypotension, parkinsonism and an increased risk of stroke and death [5, 6]. As a result, antipsychotic use is restricted to those with severe psychosis and/or aggression that have not responded to non-pharmacological approaches. In the UK, the National Institute for Health and Care Excellence guidance advocates use of 'the lowest possible dose for the shortest possible time' [7]. There is a lack of guidance on the minimum clinically effective dose for individual drugs, or the factors that predict differing response and side effects, although a recent publication, based on clinical data on risperidone use in AD, suggests that increased dementia severity is an independent risk factor for emergent parkinsonism [7].

Amisulpride is a second-generation antipsychotic–a substituted benzamide derivative which is a highly selective dopamine D2 receptor antagonist [8]. It can be prescribed off-label in very late onset (> 60 years) schizophrenia-like psychosis [9] and in patients with AD psychosis [4, 10, 11]. Therapeutic drug monitoring studies in adults predominantly aged under 65 years have shown that therapeutic striatal D2/3 receptor occupancies between 40–70% are achieved at blood drug concentrations of 100–319 ng/ml, equivalent to 400–800 mg/day [12–14]. Older patients with AD psychosis (aged 69–92 years) showed a clinical response to amisulpride and parkinsonian side effects at lower doses (25–75 mg/day) with high D2/3 receptor occupancies in the caudate (41–83%) and lower blood drug concentrations (40–100 ng/ml) than expected [10, 11].

These findings suggest that age and/or AD-related changes in central pharmacokinetics contribute to antipsychotic drug sensitivity and implicate the blood-brain barrier (BBB), which controls the entry of drugs to the brain through selective transport pathways. Evidence from our animal studies further supports this hypothesis [15]. Since amisulpride is predominantly positively [16] charged at physiological pH, observed changes in the expression and/or function of BBB transporters for organic cations may help explain the increased amisulpride sensitivity in AD patients. For example, differences in the expression of the multi-drug and toxin extrusion protein 1 (MATE1; SLC47A1) protein and the plasma membrane monoamine transporter (PMAT; SLC29A4) protein in brain capillaries isolated from healthy individuals compared to AD patients have been observed [15].

Compromised function and expression of other SLC transporters at the BBB, such as the glucose transporter, GLUT1 (SLC2A1), has also been observed in AD [17–19]. Consequently, AD brains suffer chronic shortages of energy-rich metabolites [20, 21]. Importantly, antipsychotic drugs, including risperidone and clozapine, inhibit glucose uptake by transporters and it has been suggested that this could be through direct interaction with GLUT transporters [22, 23]. In addition, the use of clozapine and another antipsychotic drug, olanzapine, has

organic cation transporters novel; NVU, neurovascular unit; PFA, paraformaldehyde; P-gp, P-glycoprotein; PHE, phenylalanine; PMAT, plasma membrane monoamine transporter; PMD, post-mortem delay; RIPA, radio-immunoprecipitation assay; SGLT1, sodium-dependent glucose transporter 1; SLC, solute carrier; TEM, transmission electron microscopy; TfR1, transferrin receptor 1; THR, threonine; TRP, tryptophan; $V_d$, volume of distribution; WB, Western blot; WT, wild type.

been associated with the development of type 2 diabetes. One of the proposed mechanisms is through the inhibition of glucose transporters, inducing hyperglycaemia [22, 24, 25]. However, detailed studies of the drugs-transporter interaction, including *in silico* molecular docking studies on the specific molecular interactions of antipsychotics with GLUT1 are rare.

Thus, considering the changes in GLUT1 expression at the BBB and resulting impact on brain metabolism in AD, and the increased sensitivity of AD patients to antipsychotic drugs including risperidone and amisulpride, we wanted to explore the interaction between GLUT1 and amisulpride. To do this we tested the hypotheses that amisulpride interacts with GLUT1 at the BBB, and that the expression of BBB transporters and the transport of amisulpride and glucose into the brain will be affected by AD. We utilised a combination of literature review, physicochemical properties analysis, molecular docking approaches, cell culture BBB studies, studies in wild type mice and in an animal model of AD and assessment of tissue from human cases with and without AD. An overview of the methods deployed are presented in S1 Fig in S1 File. Abstracts of this work have been presented [26].

## Materials and methods

### Radiolabelled and non-labelled chemicals

Radiolabelled [$^{14}$C]D-glucose (#NEC042X050UC, Lot: 2389266, specific activity 275 mCi/ mmol) and [$^{3}$H]mannitol (#NET101001MC, Lot: 3632303, specific activity 12.3 Ci/ mmol) were purchased from Perkin Elmer. [O-methyl-$^{3}$H]amisulpride (MW 374.8 g/mol; specific activity 77 Ci/mmol; 97% radiochemical purity) was custom tritiated (#TRQ41291 Quotient, UK). [$^{14}$C(U)]Sucrose (MW 359.48 g/mol; specific activity 536 mCi/mmol; 99% radiochemical purity: #MC266) was purchased from Moravek Biochemicals, USA. Non-labelled amisulpride (MW 369.5 g/mol, > 98% purity) was purchased from Cayman Chemicals, UK (#71675-85-9) and non-labelled D-glucose (#10117) was purchased from BDH, UK. Human embryonic kidney 293 (HEK293) whole cell lysate (#ab7902) and human promyelocytic leukaemia (HL-60) whole cell lysate (#ab7914) were purchased from Abcam, UK.

### Literature review

Three different groups of molecules, which interacted with GLUT, were identified by literature review. These groups were GLUT1 substrates, GLUT1 inhibitors and GLUT-interacting antipsychotics. The inclusion criteria are explained below.

**Identification of GLUT1 substrates and inhibitors.** To identify established substrates and inhibitors of GLUT1 we performed a Pubmed literature search using the parameters ((GLUT1 substrate) OR (GLUT1 inhibitor)) AND (review [Publication Type])) AND (("1985" [Date—Publication]:"2021" [Date—Publication])) [27] and tabulated the results. In the substrates group we included molecules for which there is *in vitro* evidence that they are transported by GLUT1 and that their uptake into cells is inhibited by established GLUT1 inhibitors, such as cytochalasin B. In the inhibitors group we included molecules that were shown to decrease uptake of GLUT1 substrates *in vitro* and for which there is a proposed inhibitory mechanism of interaction with GLUT1.

**Identification of antipsychotics that interact with GLUT.** We also performed a Pubmed search to identify antipsychotics which interacted with GLUT transporters using the parameters (((GLUT) AND (antipsychotic)) AND (("1985"[Date—Publication]: "2021"[Date—Publication])) (accessed 04/03/2022). In the antipsychotics group we included both typical and atypical antipsychotics that decreased the uptake of GLUT substrates *in vitro*.

## Physicochemical characterisation of GLUT1 substrates and inhibitors, and GLUT-interacting antipsychotics

The physicochemical characteristics of each member of the three groups (i.e. GLUT1 substrates, GLUT1 inhibitors and GLUT-interacting antipsychotics) were obtained from the chemical property databases, DrugBank [28] or MarvinSketch (version 22.9.0, 2022, ChemAxon) [29] and tabulated.

Specifically, information about the structure and molecular weight (MW) was obtained from DrugBank [28]. The median MW and the mean MWs of each of the three groups were then calculated and the results compared.

The gross charge distribution at pH 7.4 of each molecule was then obtained from MarvinSketch. The mean gross charge distribution at physiological pH (± SEM) of the three groups was then calculated and the results compared. The number of microspecies of each molecule at pH7.4 was examined and the percentage prevalence and charge of the top two microspecies was then reported (MarvinSketch). The physicochemical characteristics of the three groups were compared to those obtained for amisulpride to examine any overlap and identify differences.

### *In silico* molecular docking

*In silico* molecular docking was used to examine the molecular level interactions of amisulpride, alpha-D-glucose, beta-D-glucose and sucrose with GLUT1 using the molecular docking tool GOLD. We have already studied the interaction of amisulpride with several transporters (i.e. MATE1, PMAT, organic cation transporter 1 (OCT1) and P-glycoprotein (P-gp)) using molecular docking [15]. The main endogenous substrate of GLUT1 is D-glucose, which is a monosaccharide present in the body as anomers: 36% alpha-D-glucose and 64% beta-D-glucose [30]. In this molecular docking study alpha-D-glucose and beta-D-glucose were used as positive controls. Sucrose is a plant disaccharide which is thought not to interact with GLUT1 and was used as the negative control.

The GLUT1 Protein Database (PDB) code used was 5EQI–the crystal structure of human GLUT1. The 5EQI crystal structure is for the inward open conformation of the transporter which has been reported to be the most favourable for ligand binding [31].

The results from the molecular docking simulations were expressed as a free energy binding and chem score. A molecule is considered a substrate for a given transporter if their interaction has a free energy binding of ≤5kcal/mol and has a high chem score. The chem score is one of the scoring functions of the molecular docking program. It provides information on the strength of the interaction between ligand and binding sites.

### *In vitro* studies in model of the human BBB (hCMEC/D3 cells)

The human cerebral microvessel endothelial cells/D3 (hCMEC/D3) are an immortalised human adult brain endothelial cell line that stably maintains a BBB phenotype. The hCMEC/D3 cell line originated from human brain tissue obtained following surgical excision of an area from the temporal lobe of an adult female with epilepsy [32]. The surgical procedure was carried out at King's College Hospital, London, in accordance with the guidelines of the Local Ethics Committee and research governance guidelines.

The hCMEC/D3 cells are a well-established and characterised model, regularly used to study the human BBB [15, 32]. It is known to express ABC and SLC transporters including P-gp, OCT1, OCT2, OCT3, organic cation transporters novel 1 (OCTN1), OCTN2, PMAT, MATE1 and MATE2 [15, 32, 33]. This *in vitro* BBB model allowed us to: (1) study human

brain endothelium (2) confirm that GLUT1 was expressed in these cells (3) confirm the presence of a glucose transporter (4) study the interaction of amisulpride with a glucose transporter (5) minimize time-consuming animal studies in line with the framework of the replacement, reduction and refinement (3Rs) principles. To do this, the expression of a functional GLUT1 was first confirmed using Western blot (WB) and accumulation assays with [$^{14}$C]D-glucose. This was followed by further accumulation assays with [$^{14}$C]D-glucose and non-labelled amisulpride. [$^{3}$H]mannitol was used as a marker of cellular integrity and non-specific binding to the membranes and plasticware. The hCMEC/D3 cells (passages 30–35) were grown as described previously by Sekhar et al., 2019 [15].

**Western blot studies–expression of GLUT1 in hCMEC/D3 cells.**  hCMEC/D3 cells were grown for four to five days in T-75 flasks until they formed a fully confluent monolayer, and then for another two to three days before harvesting and preparing lysates for WB. Lysates were prepared by placing the flask on ice, aspirating the medium and washing the cells in ice-cold phosphate buffered saline (#70011–36 Gibco, Fischer Scientific Ltd, UK). The cells were then incubated on a shaker for 10 minutes in radio-immunoprecipitation assay (RIPA) buffer (#R0278, Sigma, UK) and 1% (v/v) protease inhibitors (#78441, Thermo Fisher Scientific, UK) for cell lysis and protein solubilisation. The cells were then scraped off from the bottom of the flask and transferred to an Eppendorf tube. The tube was then agitated for 20 to 30 minutes at 4˚C. Next, the cell suspension was centrifuged (Biofuge Fresco, Heraeus Instruments, UK) for 20 minutes at 11,753 x G at 4˚C. The resulting supernatant (protein lysate) was snap frozen in liquid nitrogen. Before the start of the WB procedure, the protein lysates were thawed and the total protein concentration in each lysate was estimated by bicinchoninic acid (BCA) assay using bovine serum albumin standards (Thermo Scientific, UK) as described by Sekhar et al., 2019 [15].

The protein lysates for WB were mixed with sample loading buffer (1:4) (#NP0007 NuPAGE LDS Sample Buffer (4x), Invitrogen, Carlsbad, USA), reducing agent (1:10) (#B0009, Novex by Life Technologies, USA), and RIPA buffer. Optimization assays were performed to compare samples heated at 95˚C for 5 minutes to samples left at room temperature [34]. The heated samples gave a stronger signal. Heating samples has previously been utilized for the Recombinant Anti-Glucose Transporter GLUT1 antibody and an anti-P-gp antibody [34, 35]. The rest of the WB procedure was performed on heated samples and as described in [15], except, 30 μg of protein was loaded in each well, antibody solutions were made in 5% milk in Tris Buffered Saline-Tween and washes were performed in Tris Buffered Saline-Tween (for antibodies dilutions see S1 Table in S1 File). Quantification of protein expression was determined by calculating the intensity ratio of the band of interest and the band of the loading control (glyceraldehyde 3-phosphate dehydrogenase; GAPDH). Band intensity ratio analysis was conducted using ImageJ software [36].

Positive and negative controls for transporter expression in hCMEC/D3 cells were chosen based on RNA-seq data from The Human Protein Atlas [37, 38]. Lysates from human colon adenocarcinoma (Caco-2) cells were used as a positive control, whilst whole cell lysate from human embryonic kidney cell line (HEK-293; (#ab7902, Abcam, UK) were used as a negative control for GLUT1 transporter expression.

**Accumulation assays–function of glucose transporters in hCMEC/D3 cells.**  The cell culture method and experimental design of the accumulation studies are described in the following sections. The cells were grown as previously described. They were split at 80–90% confluence and were seeded at 25,000 cells/cm$^2$ in the central 60 wells of 96-well plates (#10212811, Fisher Scientific, UK). They were grown for four to five days until they formed a fully confluent monolayer and then for another two to three days before the start of the accumulation assay.

For the control accumulation assay experiments, the hCMEC/D3 cells were incubated in an accumulation buffer (135 mM NaCl, 10 mM HEPES, 5.4 mM KCl, 1.5 mM $CaCl_2H_2O$, 1.2 mM $MgCl_2.H_2O$, 1.1 mM D-glucose—the standard non-labelled D-glucose concentration in the accumulation buffer, and water, pH = 7.4) for 1 h. The buffer also contained [$^{14}$C]D-glucose (3 μM) and [$^3$H]mannitol (0.05 μM). [$^3$H]Mannitol was used as a marker for non-specific binding and barrier integrity [39, 40]. All treatment conditions contained 0.05% DMSO. Incubation was performed in a shaker at 37°C and 120 rotations per minute. Next, the cells were washed with ice-cold phosphate buffered saline (#BR0014G, Oxoid Limited, England) to stop the accumulation process and to remove any radiolabelled molecules and buffer that had not entered the cells. Then the cells were incubated in 1% Triton X (200 μl per well) for 1 h at 37°C, to solubilise the cell membranes and to free any accumulated radiolabelled compounds via lysis. Half of the cell Triton X lysate from each well was pipetted into a scintillation vial and 4 ml of scintillation fluid (#6013329, Ultima GoldTM, Perkin Elmer, USA) was added. The amount of $^{14}$C and $^3$H radioactivity in the sample was measured in disintegrations per minute (dpm) on a Tricarb 2900TR liquid scintillation counter. The dpm of each sample were corrected for background dpm. The background dpm was determined from a vial containing 100 μl 1% Triton X and 4 ml liquid scintillation fluid only. The remaining cell lysate in the plate was used to measure the total protein concentration in each well, as determined by BCA analysis (17).

In our self-inhibition studies which investigated the presence of a functional GLUT1 transporter in the hCMEC/D3 cells, the cells were incubated with accumulation buffer containing [$^{14}$C]D-glucose (3 μM), [$^3$H]mannitol (0.05 μM) and 4 mM non-labelled D-glucose. The 4 mM total concentration of non-labelled D-glucose included the standard 1.1 mM non-labelled D-glucose normally present in accumulation buffer. Radioactivity uptake was compared to the control condition in which the cells were treated with accumulation buffer of the same composition apart from the additional 2.9 mM non-labelled D-glucose of the test condition. All treatments contained 0.05% DMSO. After 1 h incubation, the cells were lysed and used for liquid scintillation counting and BCA assay as already described. The non-labelled D-glucose concentration for the treatment condition (4 mM) was selected based on previous reports that normal plasma glucose concentrations are kept in a narrow range between 4 and 8 mM in healthy subjects [41] and on preliminary experiments studying the effect of 4, 6, and 9 mM non-labelled D-glucose on the membrane integrity of hCMEC/D3 cells (see S2 Fig in S1 File).

In order to assess the effect of amisulpride on the transport of D-glucose through the hCMEC/D3 membrane, non-labelled amisulpride (20, 50 or 100 μM) was added to the accumulation buffer containing [$^{14}$C]D-glucose (3 μM) and [$^3$H]mannitol (0.05 μM). All treatment conditions contained 0.05% of DMSO. The cells were incubated with this mixture for 1 h, the BCA assay and liquid scintillation counting was then performed as described previously.

The radioactivity (dpm) accumulated in the cells per mg of protein was expressed as a function of the radioactivity in a μl of the initial accumulation buffer to calculate the volume of distribution ($V_d$; μl/mg of cellular protein) of the [$^{14}$C]D-glucose or [$^3$H]mannitol as shown in the equation below.

$$Vd = \frac{dpm \; per \; mg \; of \; protein}{dpm \; per \; \mu l \; of \; accumulation \; buffer}$$

The $V_d$ for [$^{14}$C]D-glucose can be corrected with the $V_d$ for [$^3$H]mannitol (marker for non-specific binding) at each time point. Values were expressed as a mean ± SEM. The [$^3$H]mannitol values can also be used to confirm membrane integrity of each monolayer. [$^3$H]mannitol is a well-established marker of BBB integrity [42]. This method allows us to compare the

accumulation of a radiolabelled test molecule into human brain endothelial cells after a set of different experimental conditions such as self-inhibition (i.e. plus unlabelled glucose) or cross-inhibition (i.e. plus unlabelled amisulpride) studies. In this way we can explore the means of cell entry and even cell exit of the test molecules. Movement across the cell membranes may be by passive diffusion, transporter-mediated or indeed involve vesicles which may be receptor-mediated.

Statistical analysis of the results from the self-inhibition studies was performed using unpaired one-tailed Student's t-test. The results from the cross-inhibition studies were analysed using One-way ANOVA. All statistical analysis was performed with GraphPad Prism 9.0.

## Animal model studies in WT and 5xFamilial AD (FAD) mice

The 5xFAD mice are a model of AD which carry human amyloid-β precursor protein (APP) with the Swedish (K670N, M671L), Florida (I716V) and London (V717I) FAD mutations along with human PSEN-1 with two FAD mutations: M146L and L286V under the murine Thy-1 promotor [43]. This model has been described as one of the few models that shows several AD hallmarks, including neural loss, neurodegeneration, gliosis and spatial memory deficits [44]. Comparing data obtained from this mouse model of AD to that obtained from WT mice allowed us to explore the effect of disease on neurovascular structures, transporter expression in brain capillaries and on drug delivery across the BBB.

We first confirmed the model phenotype in the following ways: comparing the weight of females and males from the two genotypes; using transmission electron microscopy (TEM) to confirm the presence of amyloid plaques in the brain of 5xFAD mice; comparing the expression of APP in brain capillary enriched pellets isolated from WT and 5xFAD mice. Enrichment of the pellets with brain endothelial cells was confirmed using transferrin receptor 1 (TfR1) as an endothelial cell marker.

**Animal husbandry.** All *in vivo* experiments were performed in accordance with the Animal Scientific Procedures Act (1986) and Amendment Regulations 2012 and with consideration to the Animal Research: Reporting of *In Vivo* Experiments (ARRIVE) guidelines. The study was approved by the King's College London Animal Welfare and Ethical Review Body. The UK government home office project license number was 70/7755.

The mice were housed at King's College London in groups of three or four under standard conditions (20–22˚C, 12 h light/dark cycle) with food and water *ad libitum*. They were housed in guideline compliant cages. Animal welfare was assessed daily by animal care technicians. Animals were identified by earmarks. The experimenter was not blinded to the mouse genotype.

The WT (C57/BL6) mice and the 5xFAD mice (on C57/BL6 background) were between 4.5 and 15 months old. The average lifespan of the C57/BL6 mice has been reported to be 30 months [45]. Whereas the 5xFAD mice have been reported to have a lifespan of approximately 15 months [46], with some authors reporting a median lifespan of up to 24.6 months [45].

**TEM ultrastructural study.** To assess Aβ plaque presence in the brain in WT and 5xFAD mice we used TEM. All studied animals were female. The animals of each genotype were grouped into two age ranges. The groups were WT 4.5–6 months, 5xFAD 6 months, WT 12 months and 5xFAD 12 months with n = 2 mice per group. The weight of the animals was between 21.9 g and 26.7 g. For brain dissection, all animals were terminally anaesthetised with intraperitoneal injection of pentobarbital (100 μl/animal) (Fort Dodge, Southampton, UK).

Once each mouse was anaesthetised, it was perfuion-fixed. The left ventricle of the heart was infused with ice-cold phosphate buffered saline and the right atrium sectioned to provide an open circuit. Once the vasculature was free of blood, 4% paraformaldehyde (PFA, #F017/2,

TAAB, UK) was infused via the heart. Fixation with 4% PFA was performed for 10 minutes. The brain was then removed, and the frontal cortex was dissected and cut into 1 mm$^3$ samples.

The brain samples were processed, sectioned for TEM and then imaged. The samples were incubated overnight at 4˚C in TEM fixative containing 2.5% (v/v) glutaraldehyde in 0.1 M cacodylate buffer (pH 7.3). The tissues were post fixed in 1% (w/v) osmium tetroxide (#O021, TAAB, UK) for 1.5 h at 4˚C. Then they were washed and dehydrated through serial graded incubations in ethanol: 10%, followed by 70% and then 100%. The tissue was infiltrated in embedding resin medium (#T028, TAAB, UK) for 4 h at room temperature. The samples were embedded on flat moulds and polymerised at 70˚C for 24 h. Ultrathin sections (100–120 nm) were cut on a Leica ultramicrotome (Leica microsystems, Germany) and mounted on mesh copper grids. They were contrasted with uranyl acetate for 2 minutes and with lead citrate for 1 minute. The sections were imaged on an EM-1400 (Plus) transmission microscope operated at 120 kV (JEOL USA, Inc.).

For each frontal cortex sample, a minimum of three images were collected. Amyloid plaques were then identified and labelled with reference to an electron microscopic atlas of cells, tissues and organs [47]. The dimensions of each image were 3296x2472 pixels.

**Western blot studies.** We isolated the brain capillaries of each mouse used in the *in situ* brain perfusion experiments by capillary depletion analysis. The mouse brain was *in situ* perfused via the left ventricle of the heart with artificial plasma containing radiolabelled amisulpride and radiolabelled sucrose for 10 minutes as described below. Then it was homogenised in capillary depletion buffer (10.9 mM HEPES, 141 mM NaCl, 4 mM KCl, 2.8 mM CaCl$_2$, 1 mM MgSO$_4$.7H$_2$O, 1 mM NaH$_2$PO$_4$ and 10 mM glucose) (brain weight x 3) and 26% dextran (MW 60,000–90,000 Da) (#J14495.A1, VWR, UK) (brain weight x 4). The homogenate was centrifuged at 5,400 G for 15 min at 4˚C. This resulted in separation of an endothelial cell-enriched pellet and a supernatant containing the brain parenchyma and interstitial fluid. Half of the capillary pellets were snap frozen in liquid nitrogen and used for WB analysis to test for TfR1 and BBB transporter expression, including GLUT1, PMAT, MATE1, OCT1 and P-gp.

For WB, the mouse endothelial cell enriched capillary pellets were thawed, homogenised in 250 µl RIPA buffer with added protease inhibitor (1% v/v). The tissue was incubated in the buffer at 4˚C for 30 minutes and then centrifuged (Biofuge Fresco, Heraeus Instruments, UK) at 7,999 G for 15 minutes at 4˚C. The resulting supernatant was used for WB analysis. The rest of the WB procedure was performed as previously described in this paper. For antibodies used, see S1 Table in S1 File.

Quantification of protein expression was determined by calculating the intensity ratio of the band of interest and the band of the loading control (GAPDH or tubulin). Band intensity ratio analysis was conducted using ImageJ software [36].

Positive and negative controls for transporter expression in the mouse brain capillary lysates were chosen based on RNA-seq data from The Human Protein Atlas [37, 48]. They included mouse whole kidney lysates as positive controls for PMAT, MATE1 and P-gp expression. HEK293 was also used as a positive control for MATE1 and P-gp. Lysates from cultured human airway epithelial cells (Calu3) was used as a negative control for PMAT [37, 49]. HL-60 was used as a negative control for MATE1 and P-gp [37, 50, 51]. The positive control for OCT1 was HL-60 and negative control was Calu3. The positive control for GLUT1 expression was Caco2 cell lysates and negative control was HEK293 cell lysates.

Unpaired two-tailed Student's t-test was used for statistical analysis of the difference in expression of each transporter studied between the WT and the 5xFAD mice.

***In situ* brain perfusions.** The *in situ* brain perfusion technique allows examination of the movement of slowly moving molecule across the BBB in the absence of systemic metabolism

[15, 52]. This method was used to compare [³H]amisulpride and [¹⁴C]sucrose uptake into the brain in WT and in 5xFAD mice.

The mice were terminally anaesthetised with medetomidine hydrochloride (2 mg/kg, Veto-quinol UK Limited) and ketamine (150 mg/kg, Pfizer, UK, and Chanelle, UK), injected intra-peritoneally. Heparin was injected intraperitoneally before the perfusion (100 units heparin dissolved in 0.9% NaCl (aqueous solution) (heparin–batch number: PS40057; NaCl—Sigma Aldrich, Denmark). Two experimental groups were perfused–WT (12–15 months old, n = 7, n = 4 females, n = 3 males) and 5xFAD (12–15 months old, n = 7, n = 4 males, n = 3 females). Weights of the perfused mice were between 16.8 g and 41.5 g. Animals with a weight lower than 25 g were excluded from the perfusion analysis as perfusion at a flow rate of 5.5 ml/min could cause loss of BBB integrity [53] but their capillary lysates were used in WB experiments.

Artificial plasma was used in the perfusion experiments. It contained 117 mM NaCl, 4.7 mM KCl, 2.46 mM $MgSO_4.7H_2O$, 24.8 mM $NaHCO_3$, 1.2 mM $KH_2PO_4$, 2.5 mM $CaCl_2$, 39 g/L Dextran, 10 mM glucose, 1g/L bovine serum albumin and Evan's blue dye (0.0551 g per 1 L of artificial plasma, #E2129-10G, Sigma Life Science, India). It was warmed to 37˚C and oxygen-ated by 95% $O_2$/5%$CO_2$ gas bubbling through the solution. The artificial plasma also contained [³H]amisulpride (6.5 nM) and [¹⁴C]sucrose (9.4 μM). The artificial plasma was perfused through the left ventricle of the heart. The right atria was sectioned to allow outflow of the inflowing artificial plasma. Perfusion time was 10 minutes.

After the perfusion, the brain was dissected out and weighed. The frontal cortex, striatum, thalamus and hypothalamus were dissected under a microscope (Leica, Wetzlar, Germany), weighed and solubilised in Solvable (#6NE9100, PerkinElmer, Inc., USA) for 2–3 days, then they were taken for liquid scintillation counting. These regions were selected to compare with data from other *in situ* brain perfusion experiments and observations from human brain data sets [15] which focused on similar areas, including the caudate nucleus and the putamen, which are part of the striatum [54].

The rest of the brain was used for capillary depletion as already described. The whole brain homogenate, supernatant (containing brain parenchyma and interstitial fluid), and half of the capillary pellets were also solubilised and taken for liquid scintillation counting.

The concentration of [³H] or [¹⁴C] radioactivity present in the brain (disintegrations per minute per gram of tissue–dpm/g) was expressed as a percentage of the concentration of radio-activity detected in the artificial plasma (disintegration per minute per millilitre). The value obtained was named %Uptake showing the radioactivity in ml/g of tissue x 100. The %Uptake values for [³H]amisulpride were corrected for vascular space by subtracting the corresponding [¹⁴C]sucrose %Uptake values from the [³H]amisulpride %Uptake values.

The difference between WT and 5xFAD mice in the [³H]amisulpride uptake in the different brain regions was analysed as repeated measures data by fitting a mixed-effects model with Holm-Sidak post hoc test. The same tests were used to analyse the difference in the [¹⁴C] sucrose brain uptake between WT and 5xFAD mice. Statistical analysis of the [³H]amisulpride uptake (corrected for [¹⁴C]sucrose) into the capillary depletion analysis samples, which included the homogenate, the supernatant or the endothelial cell enriched pellet samples from WT and 5xFAD mice were analysed by unpaired Student's t-test.

## Human control and AD tissue studies

We used TEM to examine the brain and brain capillary ultrastructure of an AD case. Brain capillary depletion samples from age-matched human control and AD cases were used to eval-uate and compare the total protein levels and the expression levels of TfR1, GLUT1 and P-gp in the two groups.

**Ethics statement.** Human tissue brain samples were provided via Brains for Dementia Research (BDR) and were anonymised. Written consent was provided by BDR and the specific BDR reference numbers were: TRID_170, TRID_170 amendment, TRID_265 and TRID_287.

BDR has ethical approval granted by the National Health Service (NHS) health research authority (NRES Committee London-City & East, UK: REC reference:08/H0704/128+5. IRAS project ID:120436). Tissue samples were supplied by The Manchester Brain Bank and the London Neurodegenerative Diseases Brain Bank, both of which are part of the BDR programme, jointly funded by Alzheimer's Research UK and Alzheimer's Society. Tissue was received on the basis that it will be handled, stored, used and disposed of within the terms of the Human Tissue Act 2004. The human samples were collected during February 2012 to February 2019. The human tissue studies were conducted during 1st October 2018 to 29th January 2021. The authors conducting the human tissue analysis studies did not have access to information that could reveal the identity of the human tissue donors.

**TEM ultrastructural study.** The tissue used for TEM was from one case which was received from the London Neurodegenerative Diseases Brain Bank, Denmark Hill, King's College London. Case details: BBN002.32856; sex: female; age: 74; post-mortem delay: 19 h; pathological diagnosis: Alzheimer's disease, modified Braak staging (BrainNet Europe–BNE staging) stage VI.

Samples with size of up to 1 x 5 x 1 mm were dissected from the frontal cortex, caudate and putamen. Each sample was immersed in TEM fixative (2.5% glutaraldehyde in 0.1M cacodylate buffer) and incubated overnight at 4˚C. The samples were then post-fixed in 1% (w/v) osmium tetroxide (#O021, TAAB, UK) for 1.5 h at 4˚C. Then they were washed and dehydrated through serial graded incubations in ethanol—10% ethanol, followed by 70%, followed by 100%. The tissue was infiltrated in embedding resin medium (#T028, TAAB, UK) for 4 h at room temperature. Next, the samples were embedded on flat moulds and polymerised at 70˚C for 24 h. Ultrathin sections (100–120 nm) were cut on a Leica ultramicrotome (Leica microsystems, Germany) and mounted on mesh copper grids. They were contrasted with uranyl acetate for 2 minutes and with lead citrate for 1 minute. Finally, the sections were imaged on an EM-1400 (Plus) transmission microscope operated at 120 kV (JEOL USA, Inc.). For each section of each brain region (frontal cortex, caudate, putamen), 5 to 10 pictures were examined for pathological changes in the capillary and neurovascular unit. The original dimensions of the images are 3296 x2472 pixels.

The presence of an endothelial cell monolayer surrounded by a layer of basement membrane was considered to be an intact vessel. Capillaries could be identified in cross sectional view by their walls being composed of a single endothelial cell and their lumens containing only one to two erythrocytes. The observation of oedematous space and vacuoles around the vessels, vacuolated pericytes, large vacuoles in the endothelial cells or endoplasmic reticulum swelling, as well as oedema around the capillary and multiple layers of basement membrane were considered as signs of pathological changes [55]. Neurodegeneration was identified by the presence of lipofuscin granules, loss of myelin compactness, neurite degeneration and fibrillary deposits [56, 57].

**Western blot studies.** Post-mortem brain capillaries from healthy individuals and AD cases were used to investigate the expression of transporters. The Braak stage of the control cases was between I and II (age-related pathology only). For the AD cases, the Braak stage was between IV and VI. See Supplementary Information about the sex, age, post-mortem delay (PMD), clinical diagnosis and Braak stage of the individual cases (S2 Table in S1 File).

Brain capillaries isolated from healthy individuals and AD patients were used to investigate the expression levels of BBB transporters of interest by WB analysis. Brain capillaries were isolated after homogenising brain tissue from the frontal cortex or the caudate and carrying out a

dextran-based density-gradient centrifugation to produce a capillary enriched pellet. The capillary pellet was then homogenised in capillary depletion buffer (brain weight x 3) and 26% dextran (brain weight x 4). The homogenate was subjected to density gradient centrifugation (5,400 G for 15 minutes at 4˚C) to give an endothelial cell-enriched pellet, the resulting supernatant was discarded [58]. The pellet was further lysed in 150–200 μl ice-cold RIPA buffer with added protease inhibitors at 4˚C and then centrifuged at 8,000 G for 15 minutes at 4˚C.

The protein concentration in each lysate was determined using a BCA assay and the WB procedure was carried out as already described. For antibodies used, see S1 Table in S1 File.

We used anti-TfR1 antibodies to detect TfR1—an endothelial cell marker. This way we aimed to confirm that the capillary pellets were enriched in endothelial cells. We also used antibodies to detect GLUT1 and P-gp.

Positive and negative controls for transporter expression in the human brain capillary lysates were chosen based on RNA-seq data from The Human Protein Atlas [37, 48]. Caco-2 cell lysates were used as a positive control, and HEK293 whole cell lysate and HL-60 were used as negative controls for GLUT1 transporter expression [37, 38]. HEK-293 cell lysate was used as a positive control for P-gp transporter expression [37, 51].

Quantification of protein expression was determined by calculating the intensity ratio of the band of interest and the band of the loading control (GAPDH). Band intensity ratio analysis was conducted using ImageJ software [36].

For statistical analysis, we used two-tailed unpaired Student's t-test to compare the difference of transporter expression in the frontal cortex and the caudate between control and AD cases.

## Results

### Physicochemical characteristics of amisulpride and glucose

We determined the physicochemical characteristics of amisulpride and D-glucose using Drug-Bank and MarvinSketch. Amisulpride (chemical abstracts service (CAS) number 71675-85-9) has a MW of 369.48 g/mol, a pKa of 9.37 and exists as two microspecies at physiological pH. The predominant (96.77%) microspecies is positively charged and has a single positive charge at pH 7.4. The other microspecies (3.23%) has no charge (Fig 1). The gross charge distribution at pH 7.4 of amisulpride is +0.968.

D-glucose has a MW of 180.16 g/mol and a pKa of 11.8. In solution, at equilibrium, D-glucose exists as two anomers that interconvert spontaneously: ~36% alpha-D-glucose, and ~64% beta-D-glucose, with less than 0.01% being present as an open-chain form (linear glucose) [30, 59]. The major microspecies (99.99%) of both alpha-D-glucose and beta-D-glucose has no charge (S2 Table in S1 File). The other microspecies (0.01%) of each anomer has a single negative charge at physiological pH. The linear form of D-glucose was found to be 100% neutral. Only alpha- and beta-D-glucose have been included in the GLUT1 substrates group (S2 Table in S1 File).

### Identification of GLUT1 substrates and inhibitors

Our two PubMed searches identified 99 reviews, and 16 primary research articles, which discussed GLUT1 substrates, GLUT1 inhibitors or antipsychotics which interacted with GLUT transporters and matched our inclusion criteria. In these reviews and primary research articles, 9 GLUT1 substrates, 33 GLUT1 inhibitors and 11 GLUT-interacting antipsychotics (including 6 typical antipsychotics and 5 atypical antipsychotics) were described and are listed in S3-S5 Tables in S1 File. They include the typical antipsychotics: chlorpromazine, fluphenazine, loxapine, pimozide and spiperone. Haloperidol is also included although some studies suggest no interaction with GLUT [22, 60]. Atypical antipsychotics include: clozapine,

Amisulpride microspecies A (96.77%)          Amisulpride microspecies B (3.23%)

**Fig 1. The percentage distribution and chemical structures of the two amisulpride microspecies found at physiological pH.** Microspecies A has a single positive charge and Microspecies B has no charge, according to MarvinSketch 22.9.0.

desmethylclozapine, olanzapine, quetiapine and risperidone. S6 Table in S1 File provides specific details about the experimental evidence that indicates that these 11 antipsychotics interact with GLUT. In all cases this interaction was considered to be inhibitory.

## Physicochemical characteristics of published substrates and inhibitors of GLUT1

**Molecular weight.** The physiochemical characteristics of the identified GLUT1 substrates, GLUT1 inhibitors and the antipsychotics that interact with GLUT are summarised and compared to amisulpride (Fig 2, Table 1, and S3-S5 Tables in S1 File). The MW range for the GLUT1 substrates was 164.16 g/mol to 232.28 g/mol, with a mean±SEM of 184.95±6.45 g/mol, the MW range of the GLUT1 inhibitors was 180.16 g/mol to 518.55 g/mol with a mean±SEM of 325.50±14.4 g/mol and the MW range for GLUT interacting antipsychotics was 312.44 to 461.55 g/ mol with a mean±SEM of 369.38±16.04 g/mol (Table 1). One-way ANOVA detected differences in the MW ($F_{(2, 50)}$ = 18.72), Tukey's multiple comparisons test detected significant difference between the MW of GLUT1 substrates and inhibitors ($p<0.0001$) and between GLUT1 substrates and GLUT interacting antipsychotics ($p<0.0001$) (Fig 2A). No significant difference was observed between the MW of the GLUT1 inhibitors and the group of GLUT interacting antipsychotics (Fig 2A).

The median for the GLUT1 substrates was 180.16 g/mol, for the GLUT1 inhibitors it was 308.34 g/mol, for the GLUT interacting antipsychotics it was 375.86 g/mol (Table 1). We found that 88.89% of the GLUT1 substrates were small molecules with a MW between 101 and 200 g/ mol (Fig 2B). The GLUT1 inhibitors showed a wider range of molecular weight with the majority falling (69.7%) between 250 and 400 g/mol (Fig 2B). The majority (72.73%%) of the antipsychotics affecting GLUT1 substrates uptake were in the range of 301–400 g/mol (Fig 2B). There was no significant difference in the molecular weight between the typical (386.18±23.42) and the atypical antipsychotics (349.21±20.14) ($t$ = 1.169, df = 9, $p$ = 0.27, two-tailed unpaired t-test).

**Charge.** We investigated the predicted gross charge distribution at physiological pH of the identified substrates and inhibitors of GLUT1 as well as the GLUT interacting

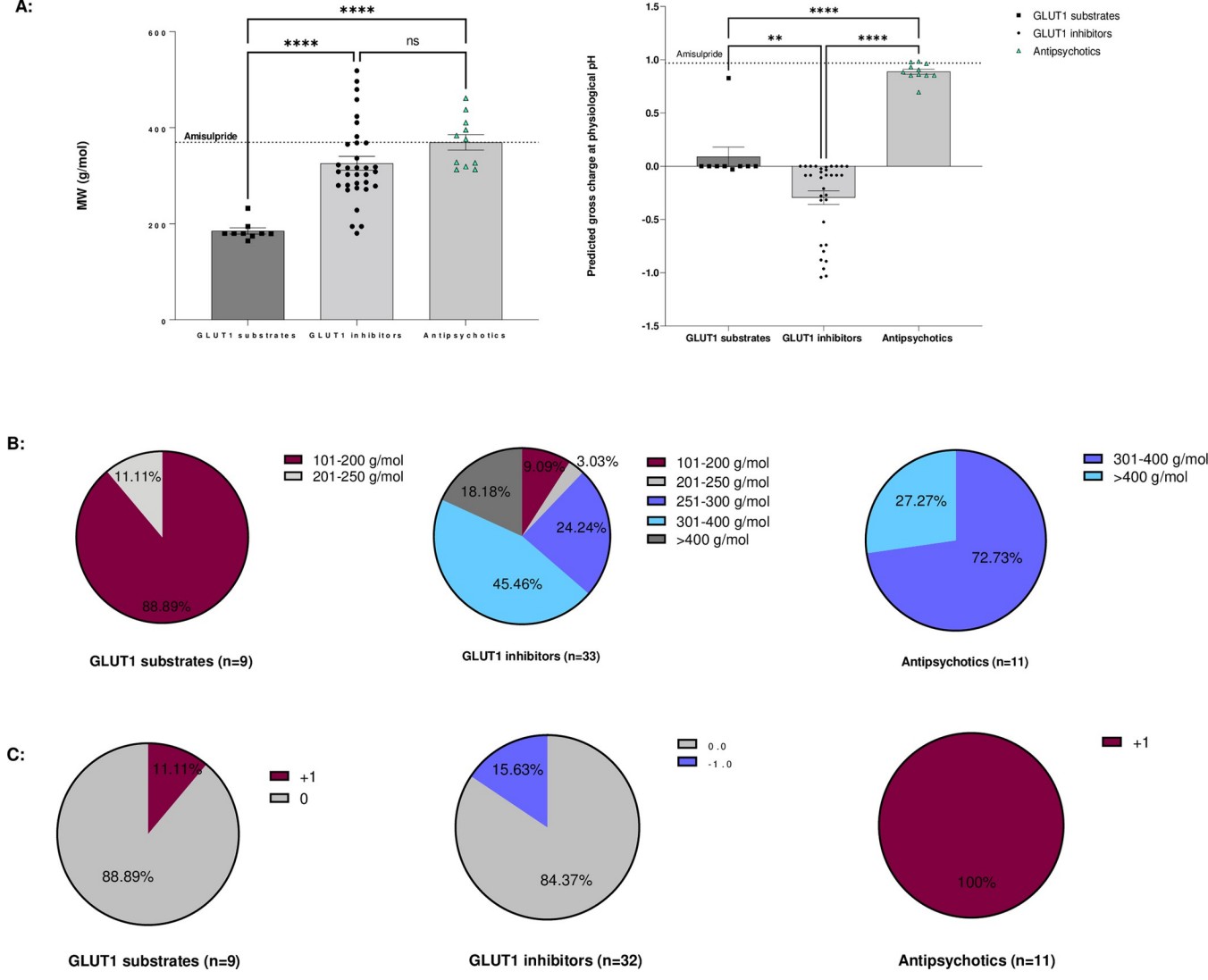

**Fig 2. Physicochemical characteristics of GLUT1 substrates and inhibitors.** Evaluation of the PubMed database identified 9 substrates and 33 inhibitors of GLUT1, and 11 antipsychotics which inhibited cell entry of GLUT substrates. A) Comparison of the molecular weight (g/mol) of GLUT1 substrates, GLUT1 inhibitors and GLUT interacting antipsychotics. Each square, dot and triangle represents a compound. Comparison of the predicted gross charge of GLUT1 substrates, GLUT1 inhibitors and GLUT interacting antipsychotics at pH = 7.4, according to MarvinSketch 22.9.0. Data was analysed using One-way ANOVA, GraphPad Prism 9 followed by Tukey's multiple comparisons test ****p<0.0001. B) The pie charts show the molecular weight of the GLUT1 substrates, GLUT1 inhibitors and G:UT-interacting antipsychotics which inhibit cell entry of GLUT substrates. Data was analysed using One-way ANOVA, GraphPad Prism 9 followed by Tukey's multiple comparisons test **p<0.01, ****p<0.0001. C) The pie charts show the charge of the most prevalent microspecies of the GLUT1 substrates, GLUT1 inhibitors, and antipsychotics interacting with GLUT at physiological pH according to MarvinSketch 22.9.0.

antipsychotics (Fig 2A), and also reported the predicted charge and the percentage distribution of the top two microspecies (S3-S5 Tables in S1 File).

Overall, we found that GLUT1 substrates, GLUT1 inhibitors and GLUT-interacting antipsychotics had an average gross charge distribution at pH 7.4 of +0.089±0.09, -0.30±0.06 and +0.89±0.02, respectively (Fig 2A and Table 1). There was a significant difference between all three groups (One-way ANOVA $F_{(2,50)}$ = 58.96, p<0.0001, Tukey's multiple comparisons test GLUT1 substrates vs GLUT1 inhibitors (p = 0.0057); GLUT1 substrates vs antipsychotics p<0.0001; GLUT1 inhibitors vs antipsychotics p<0.0001). There was no significant difference in the gross charge between the typical (0.90±0.02) and the atypical (0.87±0.05) antipsychotics (t = 0.678, df = 9, p = 0.5147, two-tailed unpaired t-test).

**Table 1. Summary of the MW range, Mean±SEM, median (g/mol) and gross charge at physiological pH of GLUT1 substrates, GLUT1 inhibitors and GLUT interacting antipsychotics.**

| | GLUT1 substrates n = 9 | GLUT1 inhibitors n = 33 | GLUT interacting Antipsychotics | | |
| --- | --- | --- | --- | --- | --- |
| | | | n = 11 | | |
| | | | All | Typical n = 6 | Atypical n = 5 |
| MW Range (g/mol) | 164.16–232.28 | 180.16–518.55 | 312.44–461.55 | 318.86–461.55 | 312.44–410.49 |
| Mean ± SEM (g/mol) | 184.95±6.45 | 325.50±14.4 | 369.38±16.04 | 386.18±23.42 | 349.21±20.14 |
| Median (g/mol) | 180.16 | 308.34 | 375.86 | 385.76 | 326.82 |
| Gross charge at physiological pH | +0.089 ± 0.09 | -0.30 ±0.06 | +0.89 ±0.02 | +0.90 ±0.02 | +0.87 ±0.05 |

Importantly, there were two molecules which could be outliers within their groups. This included the GLUT1 substrate, glucosamine, which had a much greater charge of +0.827, and the atypical antipsychotic, quetiapine, which had a lower charge of +0.696 (Fig 2A).

GLUT1 substrates existed as either one, two or three microspecies at physiological pH. GLUT1 inhibitors could exist as non-ionizable molecules (e.g. mercuric chloride) or as up to 25 microspecies (e.g. morin) at physiological pH. The antipsychotics suggested to interact with GLUT existed at physiological pH as either two, three or four microspecies. This information plus the predicted charge of the top two microspecies (if present) at physiological pH is tabulated in S3-S5 Tables in S1 File. The charge of the major microspecies of each group is presented in the form of pie charts (Fig 2C). The major microspecies of GLUT1 substrates was neutral or had +1 charge, GLUT inhibitors were neutral or had -1 charge and the anti-psychotics which interacted with GLUT all had +1 charge at physiological pH.

## *In silico* molecular docking

*In silico* molecular docking studies revealed that amisulpride could interact with GLUT1 based on the low free energy binding of the interaction, -29.04 kcal/mol and the high chem score, 26.79. The molecular docking predicted conventional hydrogen bonds between oxygen atoms from amisulpride and the amino acids: threonine (THR) A: 137 and tryptophan (TRP) A: 412, and between hydrogen atoms in the $NH_2$ group of amisulpride and asparagine (ASN) A: 411. Alkyl interaction between amisulpride and the amino isoleucine (ILE) A: 164, and pi-pi stacking interaction between amisulpride and TRP A: 412 were also observed (Fig 3A).

We also studied the interaction of GLUT1 with the monosaccharides, alpha-D-glucose and beta-D-glucose. This interaction was used as a positive control. Molecular docking found that both alpha-D-glucose and beta-D-glucose are substrates for GLUT1 with a free binding energy of -15.39 kcal/mol and chem score of 15.29. Conventional hydrogen bonds between alpha-D-glucose or beta-D-glucose, and GLUT1 were observed at TRP A: 388, glutamine (GLN) A: 282, ASN A: 411 and THR A: 137 (Fig 3B and 3C). The negative control sucrose, a disaccharide, showed a higher free binding energy and a lower chem score than amisulpride, alpha-D-glucose and beta-D-glucose. The free energy binding for the interaction of sucrose with GLUT1 was -8.58 kcal/mol and the chem score was 6.55 (Fig 3D).

## Expression and function of glucose transporters in hCMEC/D3 cells

We examined the expression and function of glucose transporters in an established cell model of the human BBB—the hCMEC/D3 immortalised cell line. Verification studies of the experimental design were also performed.

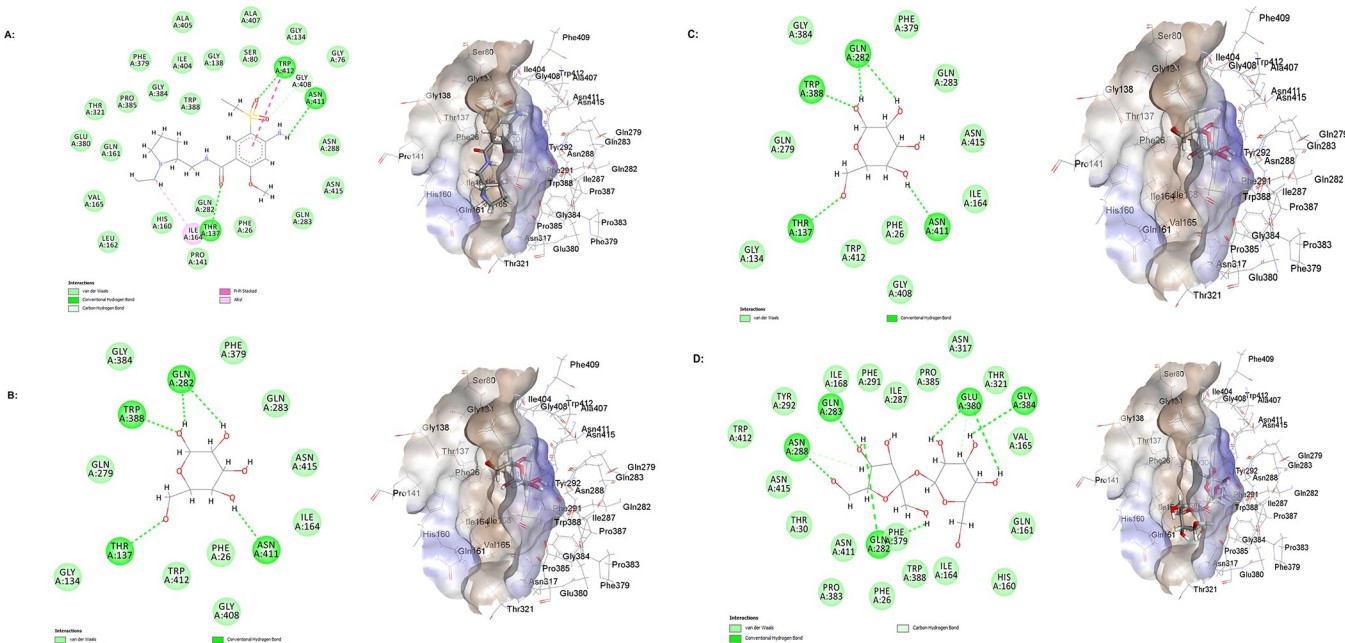

**Fig 3. Molecular level interactions of amisulpride, alpha-D-glucose, beta-D-glucose and sucrose with the binding sites of GLUT1.** 2D (left) and 3D (right) representations. In the 2D representations, green dotted lines are used to show hydrogen bonds, pink dotted lines are used to depict hydrophobic interactions. A) Stick representation is used for amisulpride and line representation for the amino acid residues from GLUT1. B) Stick representation is used for alpha-D-glucose and line representation for the amino acid residues from GLUT1. C) Stick representation is used for beta-D-glucose and line representation for the amino acid residues from GLUT1. D) Stick representation is used for sucrose and line representation for the amino acid residues from GLUT1.

**Expression of GLUT1 in hCMEC/D3 cells.**   The hCMEC/D3 cells were found to express the GLUT1 transporter (40–60 kDa). GAPDH was used as a loading control. Caco-2 cell lysates were used as positive controls and HEK293 whole cell lysates were used as negative controls (Fig 4A).

**Accumulation assays–function of glucose transporters in hCMEC/D3 cells.**   To check that excess concentrations of D-glucose did not affect the membrane integrity of the hCMEC/D3 cells, verification studies were performed. Cells were incubated with [14C]D-glucose and [3H]mannitol as a control (containing the standard amount of non-labelled D-glucose– 1.1 mM) and with [14C]D-glucose, [3H]mannitol and non-labelled D-glucose (4 mM) as a test condition. There was no significant change in the permeability of [3H]mannitol (cell permeability marker) when the cells were treated with 4 mM non-labelled D-glucose for 1 h (Fig 4B). Further analysis and studies could therefore be performed using similar D-glucose concentrations.

To investigate the function of glucose transporters in hCMEC/D3 cells, we performed a self-inhibition experiment. Incubation of the cells with 4 mM non-labelled D-glucose for 1 h led to a significant decrease in the $V_d$ of [14C]D-glucose ([3H]mannitol and protein corrected) (from 97.5 ± 22.3 µl/mg to 37.3 ±10.8 µl/mg; t = 2.161, df = 5, p = 0.0415, unpaired one-tailed Student's t-test, data is presented as mean ± SEM) (Fig 4C).

When we tested the interaction of amisulpride with glucose transporters in hCMEC/D3 cells, the entry of [3H]mannitol (0.05 µM) into the cells was used as a cell integrity marker. There was no significant change in the [3H]mannitol permeability when the cells were treated with amisulpride (20, 50 or 100 µM) (Fig 5A). There was also no significant effect on the $V_d$ of [14C]D-glucose when the cells were treated with amisulpride (20, 50 or 100 µM) (Fig 5B). Literature review confirmed that other antipsychotics inhibit glucose uptake at these

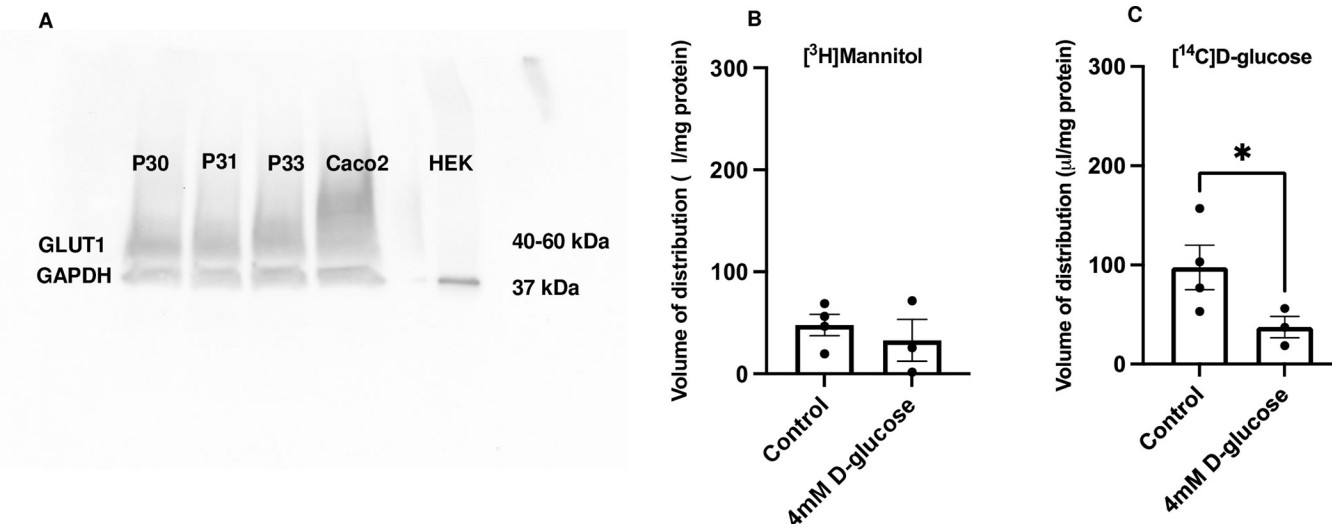

**Fig 4. Expression and function of glucose transporters in hCMEC/D3 cells–self inhibition.** A) GLUT1 expression in hCMEC/D3 cells. Three passages of hCMEC/D3 cells (P30, P31 and P33) (30 μg of protein per well) were tested for GLUT1 (40–60 kDa) expression. The figure is an example membrane of three technical repeats. Caco-2 cell lysate was used as a positive control; HEK-293 cell lysate was used as a negative control. GAPDH (37 kDa) was used as a loading control. Antibodies used: anti-GLUT1 antibody– 1:100 000, #ab115730; anti-GAPDH antibody– 1:2500, #ab9485, Abcam; secondary anti-rabbit IgG, HRP-linked antibody– 1:2000, #7074, Cell Signalling Technology. B) $V_d$ of [$^3$H]mannitol was not significantly different between the control and the experimental conditions. C) Non-labelled glucose significantly decreased the accumulation of [$^{14}$C]D-glucose ([$^3$H]mannitol corrected) in hCMEC/D3 cells after 1 h of incubation. Results are expressed as mean ± SEM, n = 3–4 plates, passages (P33, P34 and P35) with five well replicates per treatment in each plate. Data were analysed with an unpaired one-tailed Student's t-test, using GraphPad Prism 9, each point represents a plate *p<0.05.

concentrations. The half-maximal inhibitory concentration (IC$_{50}$) of these glucose-transporter inhibiting antipsychotics ranging from 2–100 μM (S6 Table in S1 File).

### Animal model of AD—validation

**Mouse weight comparison.** We compared the weight of the WT and 5xFAD mice and of the two sexes (age between 12 and 15 months). All data are presented as mean ± SEM. The WT mice had higher average weight than the 5xFAD mice (32.31 ± 2.93 g vs 26.15 ± 2.5 g) and all the 5xFAD mice with weight lower than 25 g were female. We observed that the females had significantly lower weight (g) than the males in the 5xFAD group (19.7 ± 1.46 vs 31 ± 1.61; t = 4.983, df = 5, p = 0.0042; unpaired two-tailed Student's t-test). In the WT group, weight difference between the sexes was smaller and did not reach significance (S3 Fig in S1 File).

**TEM ultrastructural study.** We examined frontal cortex samples of WT and 5xFAD mice for the presence of Aβ plaques using TEM. Structures were identified as Aβ plaques by comparison to Aβ plaques which had previously been identified in TEM images [44]. Amyloid plaques were not observed in the frontal cortex of the young WT mice (4.5–6 months; weight 25.4 g, 26.1 g). Importantly, structures identical to Aβ plaques previously reported in the literature [44] were present in the 5xFAD mice in both the young (6 months; weight 23.8 g, 23.5g) and the old (12 months 21.9 g, 22 g, data not shown) group (Fig 6).

**Western blot studies.** We performed WB for human APP to examine the genotype of each mouse used in this study. APP is the precursor protein from which amyloid-β (Aβ) is cleaved by β-secretase and γ-secretase [61]. Our studies confirmed human APP expression was not detectable in the WT mice, but was detectable in the 5xFAD mice cerebral capillaries (See S4 Fig in S1 File).

We observed TfR1 and transporter expression in the mouse brain endothelial cell lysates from WT and 5xFAD mice used in the *in situ* brain perfusions (See S5 Fig in S1 File). Thus,

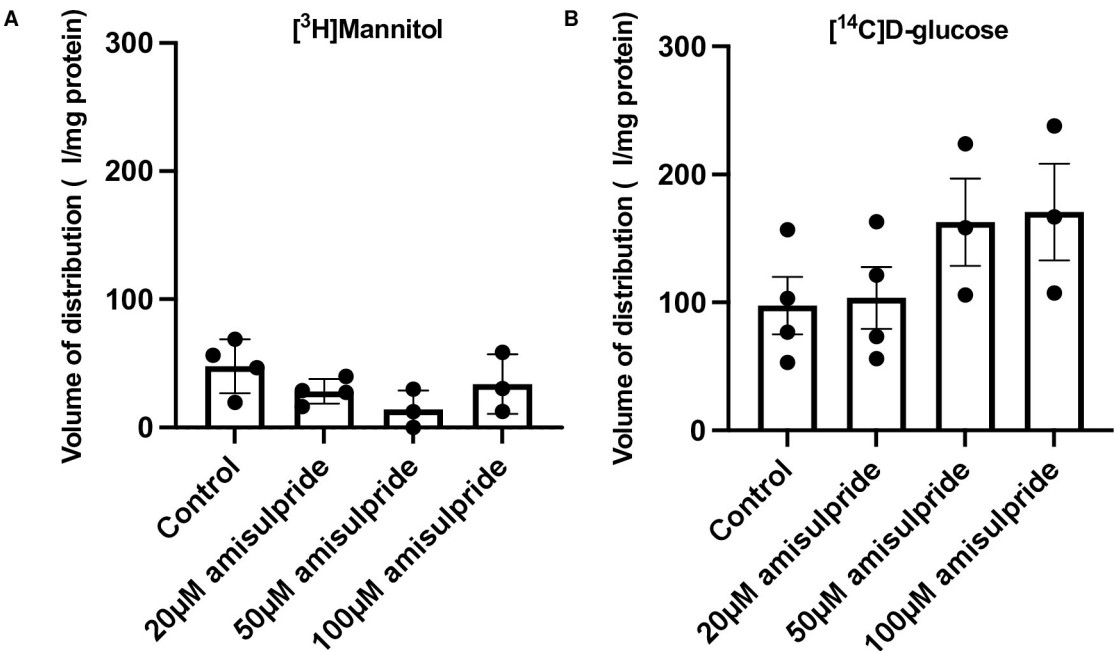

**Fig 5. Interaction of amisulpride with glucose transporters in hCMEC/D3 cells.** A) Amisulpride at three different concentrations (20, 50 or 100 μM) did not have a significant effect on the accumulation of [³H]mannitol in hCMEC/D3 cells after 1 h of incubation. B) Amisulpride at three different concentrations (20, 50 or 100 μM) did not have a significant effect on the accumulation of [¹⁴C]glucose ([³H]mannitol corrected) in hCMEC/D3 cells after 1 h of incubation. Results are expressed as mean ± SEM, n = 3–4 plates, passages (P33, P34 and P35) with five well replicates per treatment in each plate. Data were analysed with a One-way ANOVA, using GraphPad Prism 9, each point represents a plate.

these experiments confirmed we have successfully isolated the BBB compartment from the whole brain.

Brain capillaries from WT and 5xFAD mice were found to express GLUT1, PMAT, MATE1, OCT1 and P-gp. No significant effect of the genotype was found on the expression of these proteins (See S5-S9 Figs in S1 File).

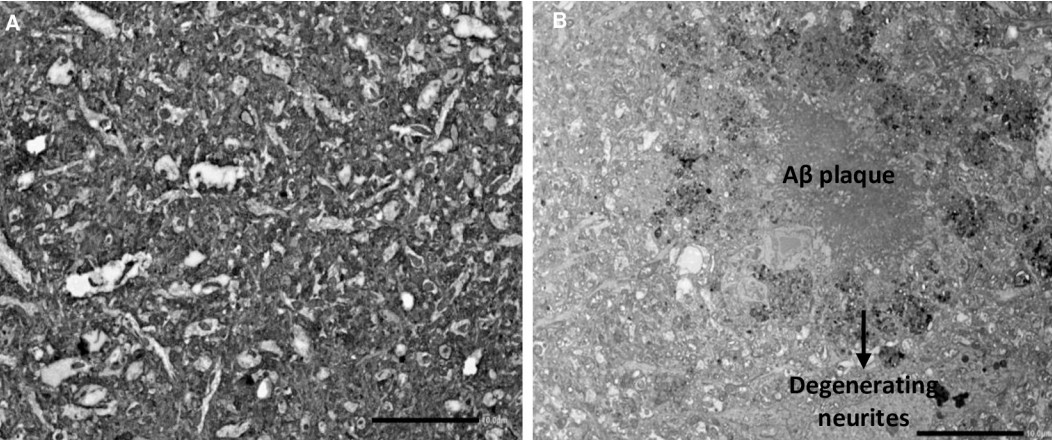

**Fig 6. TEM ultrastructural study in WT and 5xFAD mice.** TEM image of the frontal cortex of A) WT mouse, which is free of Aβ plaques and B) 5xFAD mouse, with Aβ plaques. Mouse age– 4.5–6 months. Magnification– 1200x, scale bar– 10 μm, n = 2 mice per group.

## *In situ* brain perfusions

[³H]amisulpride uptake (uncorrected for [¹⁴C]sucrose) was significantly greater than [¹⁴C] sucrose uptake in all brain areas studied apart from the hypothalamus, thalamus and capillary pellet in the WT mice (paired two-tailed t-test, data not shown). In the 5xFAD mice the [³H] amisulpride uptake (uncorrected for [¹⁴C]sucrose) was significantly greater than [¹⁴C]sucrose uptake in all brain areas studied (paired two-tailed t-test, data not shown).

When comparing WT and 5xFAD mice, there was a trend for an increase in [³H]amisulpride uptake into specific brain regions, whole brain homogenate and capillary pellet in the 5xFAD mice but this failed to reach statistical significance (Fig 7A and 7B). However, there was a statistically significant increase in the uptake of radiolabelled amisulpride in the supernatant (composed of brain parenchyma and interstitial fluid) of 5xFAD mice compared to WT mice (t = 2.550, df = 8, p = 0.0342, unpaired two-tailed t-test, by 104.86%). These results were [¹⁴C]sucrose corrected. None of the regions showed a significant difference in the [¹⁴C]sucrose uptake between the WT and the 5xFAD mice (Fig 7A and 7B).

## Human control and AD tissue studies

**TEM ultrastructural study.**   For each brain region (frontal cortex, caudate, putamen), 10 to 20 pictures were examined, and the endothelial cells, basement membranes, capillary lumens, axons, myelin sheets and markers of degeneration were identified and labelled with reference to Electron Microscopic Atlas of cells, tissues and organs [47]. We observed thickened basement membrane, vacuolisation of the endothelial cell, fibrillary deposits and oedema around brain capillaries in the frontal cortex, caudate and the putamen (Fig 8). Various other features of degeneration were observed in our AD case (BNE stage VI), including: lipofuscin granules in the frontal cortex and the putamen, and myelin degeneration in the caudate (Fig 9).

**Total protein concentration.**   A BCA assay was used to compare the total protein concentration in the frontal cortex and caudate capillary lysates of control vs AD cases. For the frontal cortex (control cases n = 9; AD cases n = 9) and caudate (control cases n = 9; AD cases n = 5), an unpaired two-tailed t-tests showed no significant difference between AD and control (S10 Fig in S1 File).

**Western blot studies.**   TfR1 was detected in the capillary protein lysates from the frontal cortex and caudate of human brains both in the control and in AD cases (Fig 10; S11 Fig in S1 File). There was no significant difference between the two groups in the frontal cortex or the caudate (unpaired two-tailed t-test). The expression of TfR1 in the samples confirms they are endothelial cell enriched (Fig 10; S11 Fig in S1 File).

We studied the expression of GLUT1 in control and AD age-matched cases (Fig 10; S12 Fig in S1 File). Human embryonic kidney 293 (HEK293) whole cell lysates (#ab7902, Abcam, UK) and/or human promyelocytic leukaemia (HL-60) whole cell lysate (#ab7914, Abcam, UK) were used as negative controls for GLUT1.

We did not see a significant change in the expression of GLUT1 between control and AD in the frontal cortex or the caudate (Fig 10).

We also studied the expression of P-gp in control and AD age-matched cases (Fig 10; S13 Fig in S1 File). The brain areas we examined were the frontal cortex and the caudate. We observed a decrease in P-gp expression in the AD group compared to control, both in the frontal cortex and the caudate. In the caudate this decrease was significant (t = 2.841, df = 16, p = 0.0118), unpaired two-tailed Student's t-test (Fig 10). Details of the specific samples used in the Western blot studies are tabulated in S7 Table in S1 File.

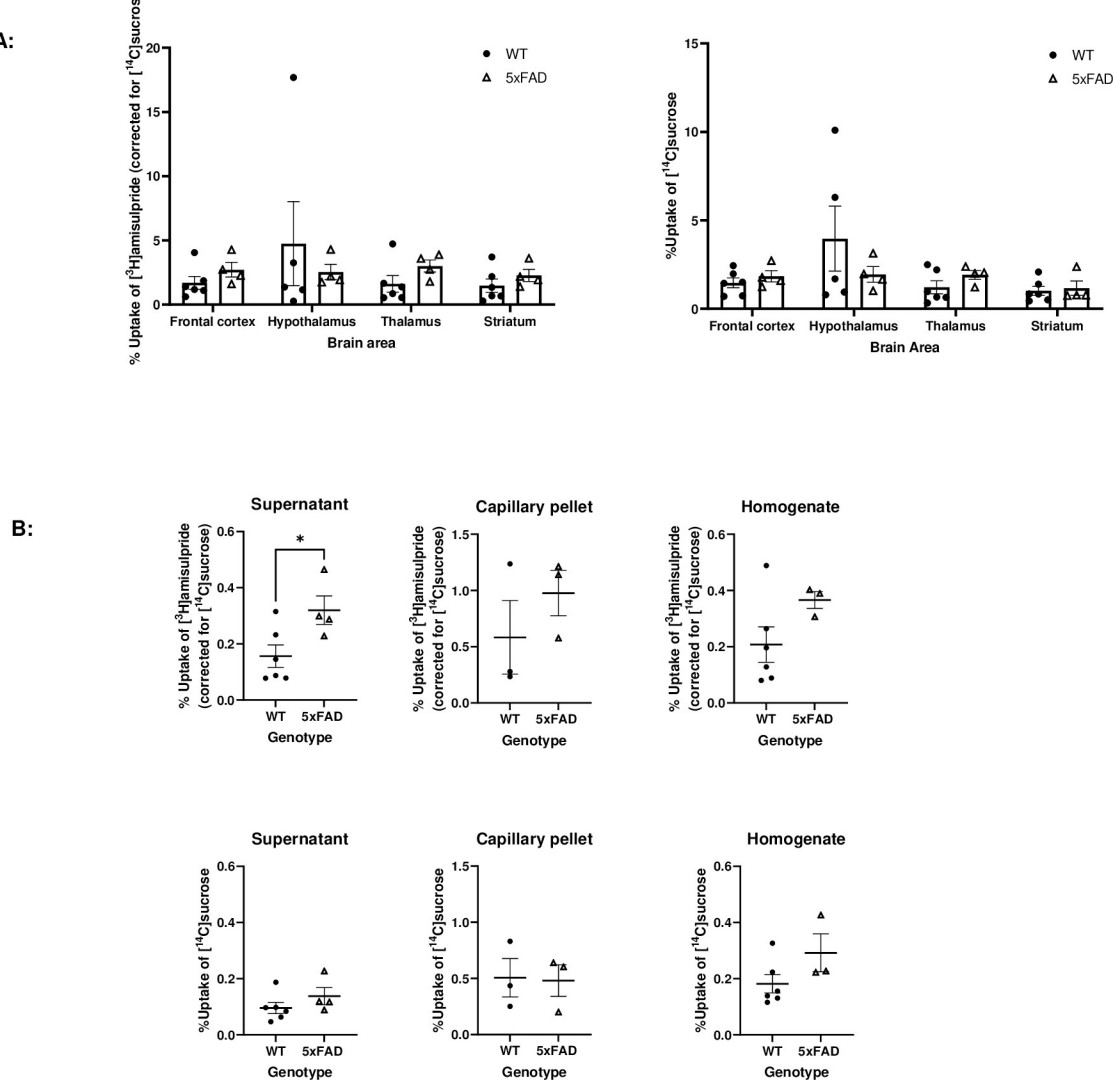

**Fig 7. [³H]Amisulpride ([¹⁴C]sucrose corrected) and [¹⁴C]sucrose uptake into the brain of WT and 5xFAD mice.** WT and 5xFAD mice were perfused with [³H]amisulpride and [¹⁴C]sucrose. Perfusion time—10 minutes, fluid flow rate– 5.5 ml/min. Mouse age: 12–15 months. WT n = 6, 5xFAD n = 4, apart for the hypothalamus where WT n = 5, 5xFAD n = 4; the capillary pellet where WT n = 3, 5xFAD n = 3, and the homogenate where WT n = 3, 5xFAD n = 3. No significant differences in paracellular permeability and membrane integrity were observed in any region. Each dot represents data from one mouse. All data are expressed as mean ± SEM. A: The effect of genotype in the different brain regions was analysed by mixed-effects analysis with Holm-Sidak post hoc test. B: The effect of genotype on the capillary depletion samples which included the whole brain homogenate, brain parenchyma containing supernatant and endothelial cell enriched pellet samples were analysed by unpaired two-tailed Student's t-test. No significant effect was observed except between the supernatant samples of 5xFAD mice compared to WT mice (*p<0.05). Statistical analysis was performed using GraphPad Prism 9.

## Discussion

This study investigated the role of GLUT1 in the transport of the antipsychotic, amisulpride at the BBB, to understand better its role in the hypersensitivity of AD patients to the side effects of antipsychotics compared to healthy aged patients [4]. We used an integrative approach to test the hypotheses that amisulpride interacts with GLUT1 at the BBB, and that expression of BBB transporters and the transport of amisulpride and glucose into the brain is affected by AD. Analysis of the published literature allowed us to identify three groups of molecules:

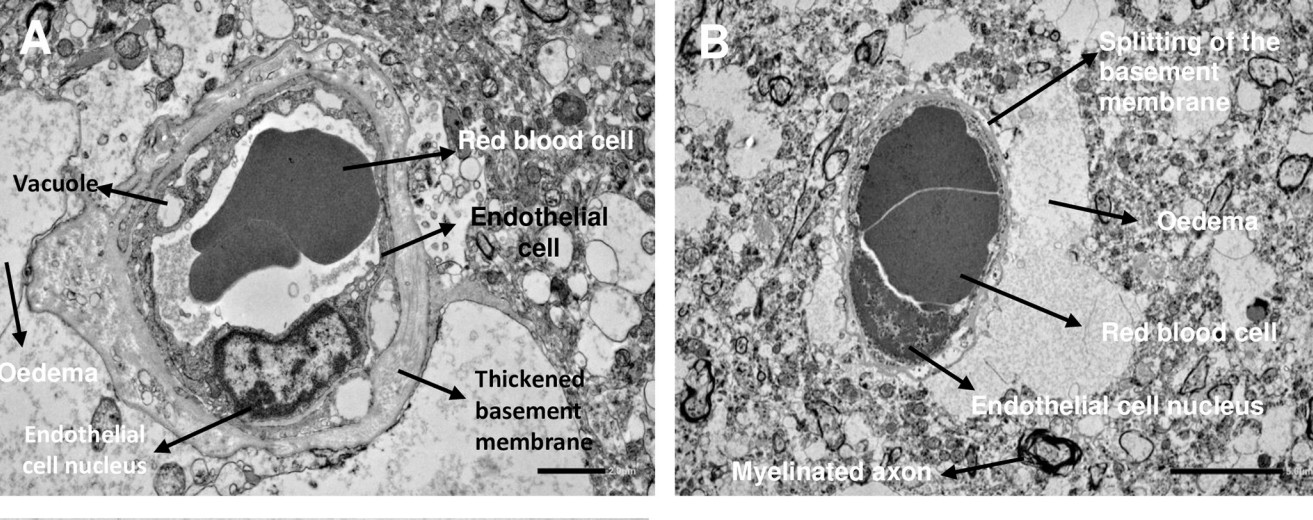

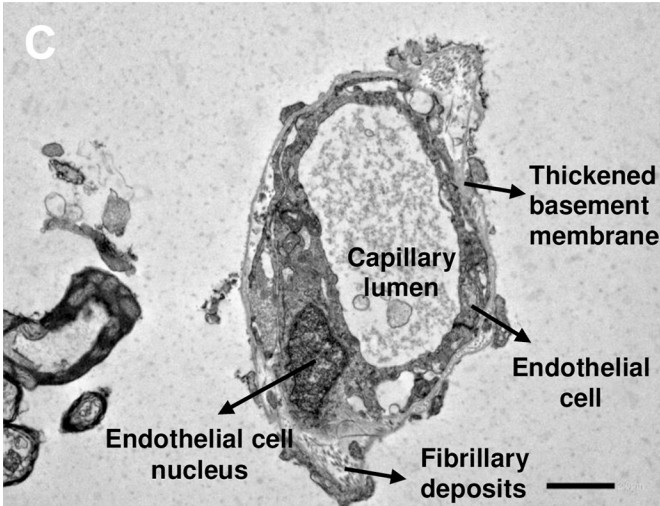

**Fig 8.** TEM image of capillaries in the frontal cortex (A) putamen (B) and caudate (C) of a human AD case. Frontal cortex and caudate magnification– 3000x scale bar– 2 μm. Putamen magnification– 2000x, scale bar– 5 μm. n = 1 case, two sections per brain area were stained and imaged, 5 to 10 images were examined per section. Case number: BBN002.32856; Sex: F; Age: 74; PM delay: 19 h; Alzheimer's disease, BNE stage VI.

GLUT1 substrates, GLUT1 inhibitors and antipsychotics that interacted with GLUT. We then utilized specialist chemical property databases to compare the physicochemical characteristics of these molecules and their groups as well as amisulpride. *In silico* docking studies allowed us to explore the specific molecular interactions of amisulpride with GLUT1. WB and *in vitro* accumulation assays in hCMEC/D3 cells were used to confirm the expression of GLUT1 protein in these BBB cells, the presence of functional glucose transporters and to determine possible interactions between glucose transporters and amisulpride. We also examined the neurovascular unit architecture in WT and 5xFAD mice at an ultrastructural level using TEM and assessed the suitability of the 5xFAD mouse model to study the BBB permeability of amisulpride in AD. WB was used to study the expression of SLC and ABC transporters, including GLUT1 at the BBB in WT and 5xFAD mice. The *in situ* brain perfusion technique allowed us to compare amisulpride uptake into compartments of the brain in WT and 5xFAD mice. We also compared transporter expression in capillaries from human cases with and without Alzheimer's dementia. We also explored the expression of SLC and ABC transporters in the same

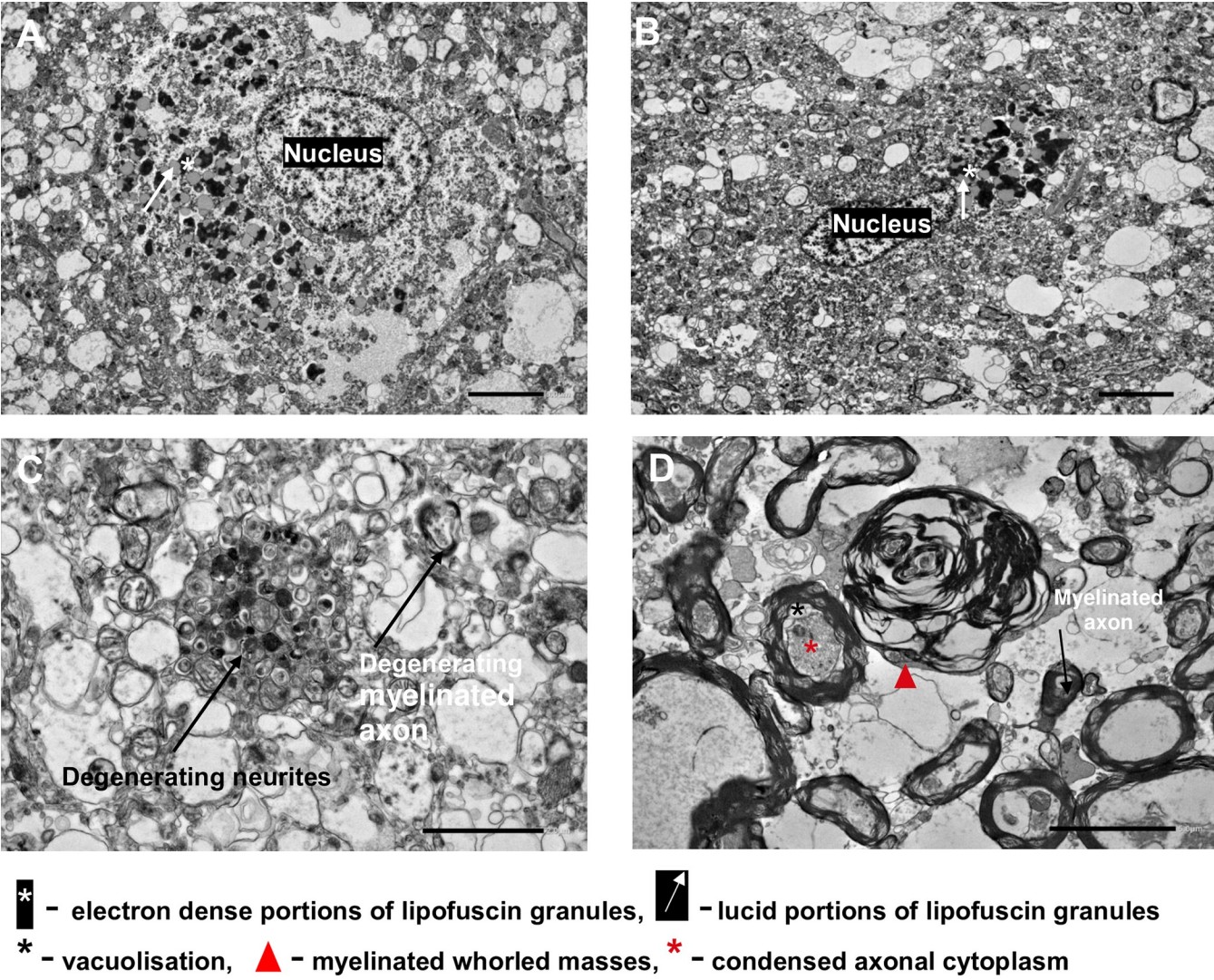

**Fig 9.** TEM image of the A) frontal cortex (A and C), putamen (B) and caudate (D) of human AD case. A) TEM image of the frontal cortex of human AD case. B) TEM image of the putamen of human AD case. A and B showing electron dense and lucid portions of lipofuscin granules. Magnification– 1500x, scale bar– 5 μm. C) TEM image of degenerating neurites and degenerating myelinated axon in the frontal cortex of a human AD case. Magnification– 6000x, scale bar– 2 μm. D) TEM image of myelin and axon degeneration in the caudate of a human AD case. Magnification– 2500x, scale bar– 5 μm. Case, n = 1, two sections per brain area were stained and imaged, 5 to 10 images were examined per section. Case number: BBN002.32856; Sex: F; Age: 74; PM delay: 19 h; Alzheimer's disease, BNE stage VI.

brain capillary sample and assessed regional differences. TEM was utilised to directly visualise cellular structures in an individual with AD.

## Physicochemical characteristics of GLUT1 substrates and inhibitors

Amisulpride is an atypical antipsychotic and has a MW of 369.48 g/mol. The main microspecies (96.77%) at physiological pH has one positive charge. The other microspecies (3.23%) of amisulpride at physiological pH has no charge. The gross charge distribution at pH 7.4 of amisulpride is +0.968.

Our literature review and analysis of the chemical property databases revealed that GLUT1 substrates are typically neutral, however, we have identified one substrate, D-glucosamine,

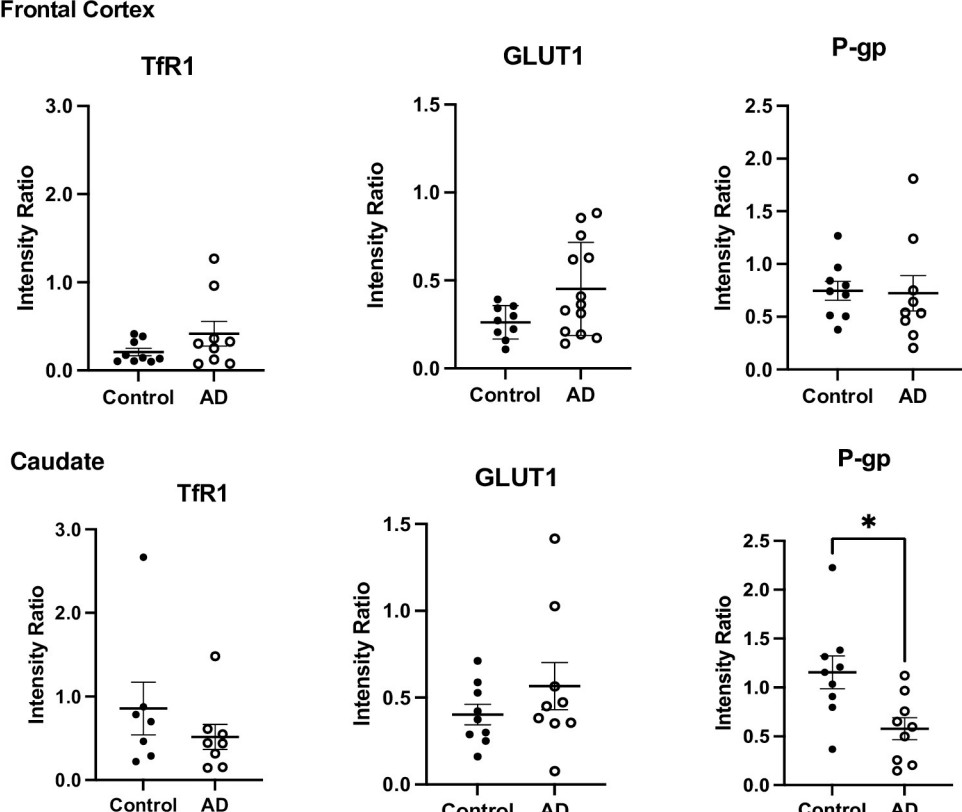

**Fig 10. TfR1, GLUT1 and P-gp expression in human frontal cortex and caudate brain capillary lysates from human control and AD age-matched cases.** The expression of the different transporters was not necessarily assessed in the same samples. The samples were not randomized. Intensity ratio was calculated using the intensity of the band of TfR1, GLUT1 or P-gp and controlling it for the intensity of the loading control GAPDH. Results are presented as mean±SEM, each dot represents a case, data was analysed using unpaired two-tailed Student's t-test. *P<0.05. All analysis was performed using Image J, Excel and GraphPad Prism 9.

which has a positive gross charge distribution physiological pH of +0.827. GLUT1 inhibitors are predominately neutral, but we identified inhibitors which had a negative gross charge distribution at pH 7.4 (for example, -1.043—Lavendustin B).

Importantly, both typical and atypical antipsychotics were identified which impede uptake of GLUT1 substrates *in vitro*. They all have a gross charge distribution which is positive at physiological pH and their main microspecies at physiological pH has a single positive charge, similar to amisulpride. An example is olanzapine for which there is also *in silico* molecular docking data for an inhibitory interaction with a bacterial glucose/H$^+$ symporter from *Staphylococcus epidermidis*. It can impede the alternating opening and closing of the substrate cavity necessary for glucose transport [62]. Another example is risperidone [23], which has been reported to interact with GLUT and to inhibit glucose uptake *in vitro* (in rat PC12 cells). Thus, it is plausible that other atypical antipsychotics such as amisulpride could interact with GLUT1.

Although it is important to consider that not all antipsychotics interact with GLUT. For example, sulpiride (CAS 23756-79-8; MW 341.43, gross charge at physiological pH +0.972) and clozapine N-oxide (a clozapine metabolite: CAS 34233-69-7; MW 342.83, gross charge at physiological pH +1.616) were reported to not have a significant effect on glucose uptake *in vitro* [22, 60].

When it comes to MW, all the established substrates of GLUT1 are smaller than amisulpride– 164.16 to 232.28 g/mol (MW of the main substrate—glucose is 180.16 g/mol). Amisulpride, has a MW similar to the established GLUT1 inhibitors and to the antipsychotics reported to inhibit uptake of GLUT1 substrates. Amisulpride is therefore more likely to represent a GLUT1 inhibitor than substrate. However, this interpretation is limited by the small number of established GLUT1 substrates, even though a large number of reviews (99) were examined.

First generation (typical) antipsychotics are known to inhibit dopaminergic neurotransmission most effectively by blocking about 72% of the D2 dopamine receptors in the brain. They also block noradrenergic, cholinergic and histaminergic receptors. Whereas, second generation (atypical) antipsychotics block D2 dopamine receptors and serotonin receptors (5-HT), mainly the 5-HT2A subtype [63]. Interestingly, amisulpride (second generation antipsychotic) shows high and similar affinities for the D2 and D3 dopamine receptor subtypes but it does not have significant affinity to the other receptor subtypes [64]. Although atypical antipsychotics are usually considered to have a broader mechanism of action compared to typical antipsychotics, there were no significant differences in physicochemical characteristics of the two groups or in their reported type of interaction with GLUT1.

### *In silico* studies

*In silico*, amisulpride was found to interact with GLUT1 with a free energy binding of -29.04 kcal/mol (the positive control, beta-D-glucose, showed free energy binding of -15.39 kcal/mol, and the negative control, sucrose, showed much higher free energy binding of -8.58 kcal/mol). Previous studies have shown that interactions with TRP412, TRP388, phenylalanine (PHE) 291, PHE379 and glutamate (GLU) 380 may play critical roles in ligand binding to GLUT1 in the inward open conformation [31]. Also, another study reported interactions between cytochalasin B, a competitive inhibitor of glucose exit via GLUT1 [65], and THR137, ASN411, GLN282, ASN288, glycine (GLY) 384 and TRP388 in GLUT1 [66]. In addition, ASN411 and TRP412 were found to be important for the binding of inhibitors in the central binding site of GLUT1 [66]. Importantly, we see interactions between amisulpride and GLUT1 at some of the reported amino acid residues–TRP412, THR137 and ASN411. The interaction of amisulpride with these specific amino acid residues in GLUT1, taken together with the size of amisulpride compared to GLUT1 substrates and inhibitors and its positive charge, suggests amisulpride might competitively inhibit glucose delivery to the CNS via GLUT1.

Dysfunction of GLUT1 is likely to cause further dysregulation of the neurovascular unit (NVU), resulting in the loss of BBB integrity as well as the observed altered transporter expression and increased Aβ toxicity in AD. Interestingly, increased neuronal uptake of glucose has been shown to protect against Aβ toxicity, and altering glucose delivery to the brain could influence progression of AD pathology [67]. A situation that may be exacerbated by the additional medications AD patients usually receive [68]. Antipsychotics are commonly prescribed with antidepressant or sedative drugs [69] i.e. citalopram [2] (MW: 324.4; +1 charge at pH 7.4) [28]. It can be assumed that if two of these psychotropics are substrates (or inhibitors) for the same transporter and are prescribed together then this is likely to change individual drug delivery and potentially nutrient delivery to the CNS and contribute to the drug hypersensitivity. In fact, polypharmacy is an important predictor of adverse drug reactions for people with and without dementia [68].

### *In vitro* studies in model of the human BBB (hCMEC/D3 cells)

We confirmed the expression of GLUT1 in the hCMEC/D3 cells with WBs. These results are in line with previous studies. GLUT1 protein expression has been detected in hCMEC/D3 cells

and in brain microvascular endothelial cells derived from healthy patient-derived induced pluripotent stem cells [70, 71]. Importantly, GLUT1 protein expression in hCMEC/D3 cells was found by quantitative proteomic analysis to be similar to that found in human brain microvessels [71]. Other glucose transporters' mRNA has also been detected in the hCMEC/D3 cells, namely GLUT3 and sodium-dependent glucose transporter 1 (SGLT1) [71, 72]. Protein expression of GLUT3 and GLUT4, but not SGLT1, has also been reported in hCMEC/D3 cells [70]. Nevertheless, there is consensus that GLUT1 is the most highly expressed transporter in the hCMEC/D3 cells and the BBB [73]. Hexose transport has previously been examined in hCMEC/D3 cells [73]. The experimental design of our accumulation assays allowed assessment of glucose transport by any glucose transporter expressed in the hCMEC/D3 cells and not just glucose transport by GLUT1.

*In vitro*, our self-inhibition study using 4 mM non-labelled glucose, showed that 4 mM glucose significantly decreased the uptake of [$^{14}$C]D-glucose into the hCMEC/D3 cells but did not affect membrane integrity as measured with [$^3$H]mannitol. This suggests the presence of functional glucose transporter(s), likely GLUT1, on the hCMEC/D3 cells.

Our *in vitro* experiments looking at the interaction between antipsychotics and glucose transport showed no effect of micromolar concentrations of amisulpride on the accumulation of [$^{14}$C]D-glucose into the hCMEC/D3 cells. This suggests that clinically relevant concentrations of amisulpride do not inhibit glucose transport (amisulpride has a plasma $C_{max}$ of 0.17–1.19 μM) [4, 74]. However, these results are difficult to interpret conclusively as the immortalized cell line may not utilize or transport glucose in the same manner as cells *in vivo*. In addition, in this *in vitro* assay, glucose transporters will be partially saturated due to the presence of non-labelled glucose in the control accumulation assay buffer. The non-labelled glucose being essential for endothelial cell survival. Although it is noted that other antipsychotic drugs have been shown to inhibit glucose uptake at these test concentrations (2–100μM) (S6 Table in S1 File).

Overall, the high affinity ($K_m$ = ~2mM [75]) of the GLUT1 transporter for glucose, high expression of GLUT1 at the BBB, excess glucose concentrations in buffer and the low concentrations of amisulpride could make the interaction between amisulpride and GLUT1 difficult to detect using the *in vitro* model of the BBB. Amisulpride use has been associated with hyperglycaemia as a side effect in 1 in 100 people [76]. The hyperglycaemia possibly being caused by inhibition of GLUT1 transport. Interestingly, recent studies did not associate its use with a higher prevalence of diabetes [77].

Cell culture studies have shown that another second-generation antipsychotic—clozapine, inhibits glucose uptake (at 20 μM) in PC12 rat cells after a 30-minute incubation. Exposure to clozapine beyond 24 h (at concentrations up to 20 μM) caused a significant increase in the cellular expression of GLUT1 and GLUT3 [22].

Later studies have confirmed that both clozapine and risperidone interact with glucose transporters and inhibit glucose transport. It has been suggested that this could be mediated by directly binding to glucose transporters and allosterically modulating either a glucose binding side or the conformational change in the protein conformation required for transport [23]. Importantly, risperidone which is the only antipsychotics licensed for use in AD, shows treatment response and emergent side effects at very low doses and low plasma concentrations [7], similar to amisulpride [4]. Also, clozapine is used at low doses and at correspondingly low plasma concentrations in the treatment of psychosis in Parkinson's disease patients [78].

## Animal model studies in WT and 5xFamilial AD (FAD) mice

In order to study the effect of AD on the BBB and how this contributes to the increased sensitivity of AD patients to antipsychotics, we need good preclinical models. In this study we

utilized the 5xFAD mice. The significantly lower weight of the female mice, compared to the male mice of the 5xFAD genotype which we observed, matches with previous reports that female mice develop the pathology earlier than male mice [79]. This could be related to increased expression of the Thy1 promoter which drives the transgenes in the 5xFAD mice and has an oestrogen response element [80] resulting in generation of higher levels of Aβ [81].

**TEM ultrastructural study.** Using TEM, we detected Aβ plaques in the frontal cortex of 5xFAD mice from 6 months of age. This is in agreement with previous data, reporting amyloid deposition in these mice from the age of 2 months [44] and with our Western blot studies which confirmed APP expression.

*In situ* **brain perfusions.** When we studied the permeability of [$^3$H]amisulpride across the BBB in a whole animal *in vivo*, we observed low permeability of the drug across the BBB in WT and in 5xFAD mice (both groups 12–15 months of age).

There was no significant change in vascular integrity, as measured by [$^{14}$C]sucrose (MW 359.48 g/mol), between age-matched WT and 5xFAD mice in all areas that we investigated even though there was a trend for increased permeability in the 5xFAD mice in the frontal cortex, thalamus, supernatant and homogenate. Interestingly, there was significantly higher uptake of [$^3$H]amisulpride (corrected for [$^{14}$C]sucrose) in the supernatant of 5xFAD mice compared to WT mice. An earlier animal study has also reported a difference in the brain uptake of amisulpride in an AD mouse model (3xTg) compared to WT, without significant change in the [$^{14}$C] sucrose uptake i.e. without significant change in BBB integrity [15]. This 3xTg mouse model of AD had a significantly increased [$^3$H]amisulpride uptake into the frontal cortex, but not into the occipital cortex, compared to WT mice at 24 months of age (age matched) [15]. This, suggests that the increased uptake of amisulpride into the brain (supernatant) of the 5xFAD mice identified in this present study could be related to regional BBB changes in transporter expression associated with AD. However, it must be mentioned that we could not detect any statistically significant changes in [$^3$H]amisulpride (corrected for [$^{14}$C]sucrose) uptake in the specific brain regions sampled from the 5xFAD mice when compared to the WT mice.

**Western blot studies.** A decrease in GLUT1 expression in brain samples has been observed previously using WB in 5xFAD mice compared to WT at 9 months of age [82]. This study used whole brain samples and pooled the samples from all of their WT mice together and from all of their 5xFAD mice together [82]. In our study we confirmed the expression of GLUT1 in 5xFAD and WT mouse capillaries at 12–15 months of age. We did not observe significant difference between the genotypes, this is likely related to the fact that we did not pool our samples but analysed each mouse separately which would have increased variability.

Other studies using the 5xFAD mouse model have reported a decrease in the expression of P-gp and GLUT1 in the cortex capillaries of 6-month-old 5xFAD mice, compared to age-matched WT mice [83]. We did not detect any effect of the genotype; this could be because we used the whole mouse brain to prepare the capillary samples and to perform WB, and the expression changes could be region-specific.

Finally, expression of PMAT, MATE1, OCT1 and P-gp has been reported in the BBB of 3xTg mice. No difference in expression levels was observed when they were compared to age matched WT mice [15]. These results are in line with our observations in the 5xFAD model. This could be because both studies used the whole brain to do capillary isolation, whereas the AD-associated transporter expression changes might be regional, as seen in human control and AD cases [15].

## Human control and AD tissue studies

**TEM ultrastructural study.** In our AD case we were able to identify brain capillaries, myelinated axons and neurites using TEM. Our images showed signs of NVU and brain

degeneration such as swollen basement membrane and oedema around the capillaries, lipofuscin granules, degenerating neurites, degenerating myelin and fibrillary depositions.

Previous EM studies have also reported abnormalities in the brain capillaries associated with AD, these include, splitting and duplication of the basement membrane, reduction of the length of the tight junctions, morphological alterations of the mitochondria of the endothelial cells, the pericytes and the perivascular astrocytic processes. The number of the pinocytic vesicles was substantially increased in the endothelium of the brain capillaries in AD in comparison with age-matched controls. Thus, it has been suggested that abnormalities in the brain capillaries may result in the release of neurotoxic factors and abnormal Aβ homeostasis in the brain and contribute to AD pathology [84].

**Western blot studies.**   Expression of the endothelial marker TfR1 in our human brain capillary lysates confirms that we have isolated brain capillary pellets enriched in endothelial cells. There was no significant change in TfR1 expression between control and AD samples observed.

Diminished glucose uptake has been reported in the hippocampus, parietotemporal cortex and/or posterior cingulate cortex in individuals at genetic risk for AD [85], positive family history [86] and/or mild or no cognitive impairment who develop AD [87, 88]. Reduced levels of GLUT1 in cerebral microvessels have also been reported in AD in the caudate nucleus [89], frontal cortex (protein was decreased but not mRNA) [18] and the hippocampus [17]. It has been suggested that GLUT1 deficiency can contribute to the disease process, acting in tandem with Aβ to initiate or amplify vascular damage and Aβ accumulation [90]. We did not observe a significant change in GLUT1 expression in the frontal cortex and caudate between controls and AD patients. This could be related to the effects of the thermal processing of the samples, which could have prevented some of the protein from entering the gel and/or production of multimers which can be mistaken for background.

P-gp is ubiquitously and abundantly expressed in the brain capillaries [91]. Importantly, protein expression has been reported to decrease significantly in the prefrontal cortex of AD patients, compared to healthy ageing controls (61 to 100 years old) [92]. In line with other studies, we observed a decrease in the P-gp expression in the caudate capillaries of AD patients, compared to healthy controls. Importantly, P-gp protects the brain from potentially toxic substances and has been reported to extrude Aβ from the brain [93]. It has been suggested that downregulation of P-gp could allow pharmaceuticals into the central nervous system and may increase the accumulation of Aβ [92].

## Conclusion

In conclusion, a literature review identified GLUT1 substrates, GLUT1 inhibitors and antipsychotics that interacted with GLUT. Physicochemical characterization of these groups using chemical property databases established that amisulpride had similar properties to the GLUT-interacting antipsychotics group. Our *in silico* molecular docking studies revealed that amisulpride interacts with GLUT1 and could potentially affect glucose delivery to the CNS. We also have *in vitro* evidence for the presence of functional glucose transporter in the hCMEC/D3 cells line. However, we could not detect any interaction of amisulpride with a glucose transporter in this assay. This is possibly because glucose being essential for endothelial cell survival limits the sensitivity of this assay for exploring antipsychotic interaction with glucose transporters *in vitro*. TEM and WB analysis validated the 5xFAD mouse model for our study. The *in situ* brain perfusion studies showed limited entry of amisulpride across the BBB in both WT and 5xFAD mice and an increased uptake into the brain of the 5xFAD mice compared to WT mice, although the cerebrovascular space was similar in both genotypes. Our WB work with P-

gp further confirms that transporter expression at the human BBB is altered in AD, although a significant difference was not observed for GLUT1 expression in our cases. It is possible that amisulpride competitively inhibits glucose entry via GLUT1, which may further compromise the neurovascular unit and increase BBB permeability. This would therefore increase central drug access, and contribute to the amisulpride sensitivity observed in AD. This research further confirms the national guidance that antipsychotic drugs should only be prescribed at the lowest dose possible for the shortest durations in AD. It is also plausible that, in the longer term, the impact on energy delivery to the brain may lead to further cellular degeneration. The implications of our findings extend to other antipsychotic drugs.

## Supporting information

**S1 File.**
(DOCX)

**S1 Raw images.**
(PDF)

## Acknowledgments

We acknowledge the support of Professor D.M. Mann from the Manchester Brain Bank who provided some of the human brain tissue samples used in this study. We would also like to thank Ms S. Selvackadunco from the London Neurodegenerative Diseases Brain Bank for her assistance with acquiring the human brain samples used in our WB and TEM studies. We also acknowledge the support of Professors I. Romero, B. Weksler and P. Couraud who provided the hCMEC/D3 cell line under MTA.

## Author Contributions

**Conceptualization:** Sevda T. Boyanova, Suzanne J. Reeves, Sarah A. Thomas.

**Data curation:** Sevda T. Boyanova, Sarah A. Thomas.

**Formal analysis:** Sevda T. Boyanova, Khondaker Miraz Rahman, Sarah A. Thomas.

**Funding acquisition:** Sarah A. Thomas.

**Investigation:** Sevda T. Boyanova, Ethlyn Lloyd-Morris, Khondaker Miraz Rahman, Doaa B. Farag, Lee K. Page, Hao Wang, Alice L. Fleckney, Sarah A. Thomas.

**Methodology:** Sevda T. Boyanova, Hao Wang, Gema Vizcay-Barrena, Roland Fleck, Sarah A. Thomas.

**Project administration:** Sevda T. Boyanova, Sarah A. Thomas.

**Resources:** Christopher Corpe, Ariana Gatt, Claire Troakes, Gema Vizcay-Barrena, Roland Fleck, Sarah A. Thomas.

**Software:** Khondaker Miraz Rahman, Doaa B. Farag.

**Supervision:** Sevda T. Boyanova, Sarah A. Thomas.

**Validation:** Sevda T. Boyanova, Sarah A. Thomas.

**Visualization:** Sevda T. Boyanova, Sarah A. Thomas.

**Writing – original draft:** Sevda T. Boyanova, Sarah A. Thomas.

**Writing – review & editing:** Sevda T. Boyanova, Christopher Corpe, Suzanne J. Reeves, Sarah A. Thomas.

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
