## [Decision Letter · Decision Letter 0]

14 Jun 2023

PONE-D-23-12121Interaction of amisulpride with GLUT1 at the blood-brain barrier. Relevance to Alzheimer’s diseasePLOS ONE

Dear Dr. Thomas,

Thank you for submitting your manuscript to PLOS ONE. After careful consideration, we feel that it has merit but does not fully meet PLOS ONE’s publication criteria as it currently stands. Therefore, we invite you to submit a revised version of the manuscript that addresses all the points raised during the review process.

Three experts have evaluated the manuscript. They found major issues to amend as described in the reviews. One of the important point is that the suitability of the model systems need to be justified.  ==============================

We look forward to receiving your revised manuscript.

Kind regards,

Mária A. Deli, M.D., Ph.D.

Academic Editor

PLOS ONE

Reviewers' comments:

Reviewer's Responses to Questions

**Comments to the Author**

1. Is the manuscript technically sound, and do the data support the conclusions?

Reviewer #1: Yes

Reviewer #2: No

Reviewer #3: Yes

2. Has the statistical analysis been performed appropriately and rigorously? 

Reviewer #1: Yes

Reviewer #2: No

Reviewer #3: Yes

3. Have the authors made all data underlying the findings in their manuscript fully available?

Reviewer #1: Yes

Reviewer #2: Yes

Reviewer #3: Yes

4. Is the manuscript presented in an intelligible fashion and written in standard English?

Reviewer #1: Yes

Reviewer #2: No

Reviewer #3: Yes

5. Review Comments to the Author

Reviewer #1: Reviewed manuscript contains very important informations for doctors, researchers and for patients with AD. Therefore, I think, this subject should be continued.

Short suggestion: For names of genes should be used italic as a font. Therefore, write please (LINE 105) name of gene (SLC2A1) according to presented suggestion.

Reviewer #2: This manuscript reports a gravely flawed analysis of the interactions of an antipsychotic drug (amisulpride) with aspects of cerebrovascular transport, including attempts at relevance to Alzheimer’s disease. The approach included literature review to identify and characterize confirmed and proposed antipsychotics that interact with glucose transporter 1 (GLUT1), in-silico modeling of amisulpride with GLUT1, in-vitro tests of glucose uptake in a transformed endothelial cell line, and a in-vivo perfusion assay of amisulpride uptake into the brain of a mouse model of amyloid accumulation. Levels of some transport enzymes (including GLUT1) in mouse and human brain were also assayed, and the ultrastructural integrity of cerebrovascular structure was evaluated in an Alzheimer’s case. Despite the lack of physicochemical data to this effect, the authors conclude that amisulpride may alter glucose uptake by GLUT1 and that this necessarily applies to other antipsychotic drugs. They also conclude that levels of GLUT1 are not altered in AD or a mouse model of amyloid accumulation and that P-gp levels are lower in AD brains but only in the caudate.

The perfusion approach used to assess amisulpride uptake is unusual and likely inappropriate. Similar assays of GLUT1 transport of glucose or its analogs is performed on awake, unanesthetized animals because anesthetics interfere with uptake and/or metabolism. Here, mice were anesthetized with medetomidine and ketamine. Ketamine has been shown to have direct effects on GLUT1 (PMID: 10927023). Medetomidine is an alpha-adrenergic R2 agonist, and these receptors have dramatic effects on glucose regulation. Whether or not there is a more proximal effect of medetomidine on GLUT1, the increase in blood glucose levels that results from alpha-adrenergic R2 agonism would certainly make an appreciable impact on GLUT1 function. Incidentally, it is never stated where was the perfusion was introduced; one assumes it was performed transcardially, as was the perfusion for fixation.

The hCMEC/D3 cell line is not a valid model for glucose uptake by cerebrovascular endothelium. These cells have been immortalized by viral transduction of telomerase (hTERT) and SV40 large T antigen. This has almost certainly altered their glucose metabolism and transport (PMID: 2998464, PMID: 1316810, PMID: 11390181). Among other alterations, hCMEC/D3 cells express considerable amounts of GLUT3 and SGLT1, which are not present in normal endothelial cells, in mouse at least. If they are significantly expressed normal human cerebrovascular endothelium, this would behoove the authors to include these proteins in their analysis of AD versus control.

TEM analysis of vascular morphology was performed on only one human sample (AD).

Why was glucose uptake represented in units of volume? How can this make sense? It seems to assume that cells import glucose by pinocytosis of the buffer or medium surrounding them.

Suppl. Fig. 4 is troubling. Many of the 5xFAD mice appear to lack APP, as does the “positive control” 3xTg. Some of the samples are even inconsistent between the two gels: M4 and M5 have no APP in the first membrane but robust levels in the second, and M3 is the opposite. It is very difficult to understand how the authors can claim “WB analysis validated the 5xFAD mouse model for our study.” [Lines 1126–1127]

It is inappropriate to heat samples to 95 deg-C before western-blot analysis of membrane transport enzymes. Such high heat causes stable aggregation of GLUT1 and P-gp, retarding entry of a large fraction of these proteins into polyacrylamide gels. Suppl. Figs. 6 and 12 makes it clear that GLUT1 was not successfully detected or quantified in this study, rendering the “data” in Fig. 10 and Suppl. Fig. 9 meaningless.

Suppl. Table 2 indicates 9 and 14 samples were available for controls and AD, respectively. But not all of these were analyzed for every protein target. How did the authors select those which were omitted? Was there randomization?

A statistician should be consulted regarding the use of t tests for analyses that include several brain regions or other variables that result in multiple measurements (e.g., Figs. 7 and 10). Most statisticians would caution against this due to the introduction of risk for Type-1 error.

Interpretations/conclusions

“This suggests that antipsychotics exacerbate the cerebral hypometabolism in AD.” This is not a logical inference from the data. An impact of amisulpride on glucose uptake was not demonstrated.

Despite the fact that a literature review was a highlighted part of this report, an enormous body of literature regarding glucose transporters and P-gp in AD and mouse models of amyloid accumulation was ignored in the Discussion.

In the authors’ prior work (PMID: 31842924), amisulpride was ruled out as a P-gp substrate. Thus, it is odd that P-gp was a molecule of some intensive focus here.

Language usage

Much of the language usage is unsophisticated or irregular.

Lines 44–45: “There was no significant effect of AD on mouse GLUT1 and P-gp BBB expression….” Of course, there is not; mice do not get AD. It well-recognized in the AD field that claiming (or implying) that transgenic models of amyloid accumulation are models of AD is a serious misrepresentation.

There is a strange insertion of commas in many places that are completely uncalled for. Some of these convey an incorrect meaning to the sentences in which they appear. For instance, a comma should only separate a phrase from the item it follows if it is a nonessential phrase, i.e. one that applies to all such items. Thus, in Lines 83–84, “Amisulpride is a second-generation antipsychotic – a substituted benzamide derivative, which is a highly selective dopamine D2 receptor antagonist,” the comma indicates that all substituted benzamide derivatives are highly selective D2R antagonists. That, of course, is not true. The authors are attempting to convey that amisulpride is special among substituted benzamides in its ability to antagonize D2Rs, and that requires the absence of a comma after “derivative.”

The complementary error is made in the sentence that appears in Lines 94–97. Here, comma is required after “(BBB).” Otherwise, the meaning conveyed is that there is a special subset of the BBB which controls the entry of drugs into the brain, whereas the truth of the matter is that the entire BBB contributes to this restriction. Likewise, the omission of a comma in Lines 111–112 leaves the impression that there is a class of glucose transporters that serve to induce hyperglycemia.

In addition, use of the “Oxford comma” is inconsistent.

Semicolons are also used unnecessarily, as in Line 88, where a comma would be appropriate.

In Lines 100–102, an incomplete sentence is inserted.

Line 211: “plastic wear” should be “plasticware.”

What does the “self” in “self-inhibition studies” mean? Glucose? If it is unlabeled glucose, it is not “self” with respect to tritiated glucose. This is strange terminology for a cold-competition experiment.

Lines 276–277: The following is odd: “The 4 mM total concentration of non-labelled D-glucose included the standard 1.1mM non-labelled D-glucose normally present in accumulation buffer.” Are the authors attempting to convey that they did not add an additional 4 mM glucose on top of glucose that pre-existed in the buffer at 1.1 mM? Why would anyone assume that they did? The meaning of “total” should be clear to any reader.

The sentence that follows that one is also odd: “Radioactivity uptake was compared to the control condition in which the cells were treated with accumulation buffer of the same composition apart from the non-labelled D-glucose content which was 1.1 mM (the standard non-labelled D-glucose concentration in the accumulation buffer).” Is this to say that the control was exactly the same as the experimental treatments except that the concentration of non-labelled glucose was lower by 1.1 mM (i.e., totaled 2.9 mM)? If they are attempting to convey that the control condition was identical except that the concentration of non-labelled glucose was 1.1 mM (i.e., lowered by 2.9 mM), a comma would be needed after “content.” Regardless, it is not clear how this is a control. Is it meant to be a positive control under the assumption that more radiolabeled glucose would be taken up when the unlabeled competitor was at a lower concentration? This is a very strange way to describe a cold-competition experiment. Moreover, a 3.6-fold elevation of cold competitor is hardly enough to achieve a blockade of uptake.

The formatting of the manuscript was very unusual. While the figures were compiled at the end, figure legends were interspersed throughout the main body of the text. This made the paper rather difficult to read/evaluate.

Reviewer #3: The manuscript titled "Interaction of amisulpride with GLUT1 at the blood-brain barrier. Relevance to

Alzheimer’s disease" by Boyanova and colleagues investigated the possible interactions of amisulpride (an antipsychotic). Overall, it is a very good manuscript but has some short-term.

Major issue: Fig.4A. I am not convinced by the blot, especially capable to run GLUT1 and GAPDH on the same blot at such a low resolution, but also the absence of GLUT1 band in HEK293 cells. HEK293 cells express a non-negligible amount of GLUT1, which suspects that the antibody is not suited or not good enough. I recommend to perform this blot with SA0377, a Rb monoclonal GLUT1 antibody targeting the C-terminal fraction of the transporter.

Minor issue: Please show the data as Mean+SD, also provide the P-value if P<0.10 so the reader can assess is there is a trend or not to significance, and if the authors have the Km (substrate) and IC50 (inhibitor) for GLUT1 of the compounds listed. These can help the reader appreciated the selectivity and affinity of these compounds to GLUT1.

6. PLOS authors have the option to publish the peer review history of their article (what does this mean?). If published, this will include your full peer review and any attached files.

Reviewer #1: No

Reviewer #2: No

Reviewer #3: **Yes: **Abraham J. Alahmad

---

## [Author Response · Author response to Decision Letter 0]

10 Jul 2023

We would firstly like to take this opportunity to thank the reviewers for their constructive comments. 

Q1: The reviewers mentioned that the suitability of the model systems should be justified.

IN RESPONSE: 

We have now provided further justification of the methods used in the manuscript.

In this manuscript multiple approaches were used to test the hypotheses that amisulpride interacts with GLUT1 at the BBB, and that the expression of BBB transporters and the transport of amisulpride and glucose into the brain will be affected by AD. 

As stated on line 136: 

‘We utilised a combination of literature review, physicochemical properties analysis, molecular docking approaches, cell culture BBB studies, studies in wild type mice and in an animal model of AD and assessment of tissue from human cases with and without AD. An overview of the methods deployed are presented in Supplementary Fig 1, S1 File’. 

Literature review

As stated on line 155:

‘Three different groups of molecules, which interacted with GLUT, were identified by literature review. These groups were GLUT1 substrates, GLUT1 inhibitors and GLUT-interacting antipsychotics’. 

In fact we were able to identify: 

As stated on line 652:

9 GLUT1 substrates, 33 GLUT1 inhibitors, and 11 GLUT-interacting antipsychotics

Physicochemical properties analysis

This was followed by physicochemical characterization of the three groups using specialist chemical property databases. 

We have now added more information to line 194:

‘The physicochemical characteristics of the three groups were compared to those obtained for amisulpride to examine any overlap and identify differences’.

This information provided the basis for the in silico, in vitro and in vivo experimental design described below.

In silico approaches

In silico molecular docking studies allowed us to explore the specific molecular interactions of amisulpride, alpha-D-glucose, beta-D-glucose, and sucrose with GLUT1. 

We have now further justified the in silico models by adding the following sentence to the manuscript: Line 200:

‘We have already studied the interaction of amisulpride with several transporters (e.g. MATE1, PMAT, organic cation transporter 1 (OCT1) and P-gp) using molecular docking (Sekhar et al., 2019).’

Cell culture models

We have now further justified the cell culture models by adding the following information to the manuscript: 

Line 221:

The human cerebral microvessel endothelial cells/D3 (hCMEC/D3) are an immortalised human adult brain endothelial cell line that stably maintains a BBB phenotype. 

We have also added the following sentence Line 223:

‘It is known to express ABC and SLC transporters including P-gp, OCT1, OCT2, OCT3, organic cation transporters novel 1 (OCTN1), OCTN2, PMAT, MATE1 and MATE2 (Sekhar et al., 2017; 2019)’. 

We have also added the following information to line 227:

This in vitro method allowed us to: (1) study human brain endothelium (2) confirm that GLUT1 was expressed in these cells (3) confirm presence of a glucose transporter (4) study the interaction of amisulpride with a glucose transporter (5) minimize time-consuming animal studies in line with the framework of the replacement, reduction and refinement (3Rs) principles. 

Animal model of AD and WT

We have added the following information to line 369:

Comparing data obtained from this mouse model of AD to that obtained from WT mice allowed us to explore the effect of disease on neurovascular structures, transporter expression in brain capillaries and on drug delivery across the BBB. 

Human tissue 

We have added the following sentence to the manuscript to further justify use of human tissue: Line 1009:

‘We also explored the expression of SLC and ABC transporters in the same brain capillary sample and assessed regional differences.

Our responses to other comments are listed below.

Q2. Reviewer #1: Reviewed manuscript contains very important informations for doctors, researchers and for patients with AD. Therefore, I think, this subject should be continued. Short suggestion: For names of genes should be used italic as a font. Therefore, write please (LINE 105) name of gene (SLC2A1) according to presented suggestion.

Q2: RESPONSE: We have reviewed each sentence which includes an SLC code to determine if we were referring to the protein or the gene. 

We specified a specific SLC code three times (Lines 108, 109 and 113). 

On line 108 we specified SLC47A1 and on line 109 we specified SLC29A4. 

The reference we cited was:

16. Sekhar GN, Fleckney AL, Boyanova ST, Rupawala H, Lo R, Wang H, et al. Region-specific blood–brain barrier transporter changes leads to increased sensitivity to amisulpride in Alzheimer’s disease. Fluids Barriers CNS. 2019 Dec 17;16(1):38.

We re-examined the reference and, in both instances, we were referring to the protein code so the SLCs should not be italized. We have now confirmed this point in these sentences by addition of the word: protein.

On line 113, we specified SLC2A1. We re-examined the three papers cited. References 18-20.

18. Horwood N, Davies DC. Immunolabelling of hippocampal microvessel glucose transporter protein is reduced in Alzheimer’s disease. Virchows Arch. 1994;425(1):69–72. PROTEIN

19. Mooradian AD, Chung HC, Shah GN. GLUT-1 expression in the cerebra of patients with Alzheimer’s disease. Neurobiol Aging. 18(5):469–74. PROTEIN

20. Kalaria RN, Harik SI. Reduced glucose transporter at the blood-brain barrier and in cerebral cortex in Alzheimer disease. J Neurochem. 1989 Oct;53(4):1083–8. 

In all cases the changes that were observed were because of protein changes or changes in function. So the SLC code should not be italized.

Q3. Reviewer #2: This manuscript reports a gravely flawed analysis of the interactions of an antipsychotic drug (amisulpride) with aspects of cerebrovascular transport, including attempts at relevance to Alzheimer’s disease. The approach included literature review to identify and characterize confirmed and proposed antipsychotics that interact with glucose transporter 1 (GLUT1), in-silico modeling of amisulpride with GLUT1, in-vitro tests of glucose uptake in a transformed endothelial cell line, and a in-vivo perfusion assay of amisulpride uptake into the brain of a mouse model of amyloid accumulation. Levels of some transport enzymes (including GLUT1) in mouse and human brain were also assayed, and the ultrastructural integrity of cerebrovascular structure was evaluated in an Alzheimer’s case. Despite the lack of physicochemical data to this effect, the authors conclude that amisulpride may alter glucose uptake by GLUT1 and that this necessarily applies to other antipsychotic drugs. They also conclude that levels of GLUT1 are not altered in AD or a mouse model of amyloid accumulation and that P-gp levels are lower in AD brains but only in the caudate.

Q3 RESPONSE: The reviewers summary of the manuscript is incorrect. 

Firstly, we are not saying that amisulpride may alter glucose uptake by GLUT1 and that this applies to other antipsychotic drugs.

We have in silico evidence that amisulpride interacts with GLUT1 and other groups have evidence that other antipsychotic drugs interact with glucose transporters including GLUT1. In fact, 11 antipsychotics were identified which interact with glucose transporters (see Supplementary Table 5, S1 File).

So there is evidence from other sources that would suggest amisulpride has potential to interact with GLUT. 

Secondly, we also did not assess ‘enzymes’ we assessed the expression of ‘transporters and receptors’.

Thirdly, the physicochemical characterization data does not allow us to rule out that amisulpride interacts with GLUT.

Regarding the physicochemical characteristics:

-The +1 charge of amisulpride is similar to the other antipsychotics that interact with the glucose transporters. 

-In addition, we identified one GLUT1 substrate that has a +1 charge so there is evidence that other types of molecules that interact with GLUT 1 have a similar charge.

-The MW of amisulpride suggested it would more likely be an inhibitor of GLUT1 transport than a GLUT1 substrate due to its similarity to other GLUT1 inhibitors.

Q4: The perfusion approach used to assess amisulpride uptake is unusual and likely inappropriate. Similar assays of GLUT1 transport of glucose or its analogs is performed on awake, unanesthetized animals because anesthetics interfere with uptake and/or metabolism. Here, mice were anesthetized with medetomidine and ketamine. Ketamine has been shown to have direct effects on GLUT1 (PMID: 10927023). Medetomidine is an alpha-adrenergic R2 agonist, and these receptors have dramatic effects on glucose regulation. Whether or not there is a more proximal effect of medetomidine on GLUT1, the increase in blood glucose levels that results from alpha-adrenergic R2 agonism would certainly make an appreciable impact on GLUT1 function. Incidentally, it is never stated where was the perfusion was introduced; one assumes it was performed transcardially, as was the perfusion for fixation.

Q4: RESPONSE:

Anaesthetic use: The reviewer is concerned that the anaesthetics will alter the glucose concentration of the plasma. However, the plasma used in the perfusion assay is artificial and as a result the concentration of glucose in the plasma is stable throughout the experiment.

Specify route of perfusion: The reviewer mentioned we had not specified the route of perfusion.

On line 450 we state that the: ‘The mouse brain was in situ perfused via the heart with artificial plasma.’

However, we have now included an additional sentence to explain this in more detail:

On line 499: ‘The artificial plasma was perfused through the left ventricle of the heart. The right atria was sectioned to allow outflow of the inflowing artificial plasma’.

Q5: The hCMEC/D3 cell line is not a valid model for glucose uptake by cerebrovascular endothelium. These cells have been immortalized by viral transduction of telomerase (hTERT) and SV40 large T antigen. This has almost certainly altered their glucose metabolism and transport (PMID: 2998464, PMID: 1316810, PMID: 11390181). Among other alterations, hCMEC/D3 cells express considerable amounts of GLUT3 and SGLT1, which are not present in normal endothelial cells, in mouse at least. If they are significantly expressed normal human cerebrovascular endothelium, this would behoove the authors to include these proteins in their analysis of AD versus control.

Q5: RESPONSE:

5a) We would like to highlight the following paragraph to the reviewers: 

Please note the bolded sentences.

Lines 1103-onwards.

‘We confirmed the expression of GLUT1 in the hCMEC/D3 cells with WBs. These results are in line with previous studies. GLUT1 protein expression has been detected in hCMEC/D3 cells and in brain microvascular endothelial cells derived from healthy patient-derived induced pluripotent stem cells (61,62). Other glucose transporters’ mRNA has also been detected in the hCMEC/D3 cells, namely GLUT3 and sodium-dependent glucose transporter 1 (SGLT1) (62,63). Protein expression of GLUT3 and GLUT4, but not SGLT1, has also been reported in hCMEC/D3 cells (61). Nevertheless, there is consensus that GLUT1 is the most highly expressed transporter in the hCMEC/D3 cells and the BBB (64).’

5b) Line 1106: We have now added that: 

-‘Importantly, GLUT-1 protein expression in hCMEC/D3 cells was found by quantitative proteomic analysis to be similar to that found in human brain microvessels (Ohtsuki et al., 2013)’. 

Line 1113: We have now added that:

-Hexose transport has previously been examined in hCMEC/D3 cells (Meireles et al., 2013).

5c) We have also been more general when describing the in vitro hCMEC/D3 accumulation assay. So we do not specify GLUT1 transport exactly we soften the statements to say glucose transport or glucose transporters e.g. Lines 282, 773,774, 784, 804, 814, 822, 831, 1001, 1002, 1114, 1123, 1126, 1128, 1280 and 1282. 

This clarifies that the experimental design allows assessment of glucose transport by any glucose transporter expressed in the hCMEC/D3 and not just glucose transport by GLUT1. We have specifically stated this now on Lines 1114-1116.

Q6: TEM analysis of vascular morphology was performed on only one human sample (AD).

Q6 RESPONSE: Data from human tissue samples are rare and it is not a simple process to study these precious samples. In particular, samples for TEM must have a very brief post-mortem delay (i.e. up to 22 h) and be processed for TEM in a specific manner. 

We were unable to obtain any further samples which had a PMD less than 22 hours in the time frame of this study. However, we considered that the importance and rarity of the images from the one donor that we did have, merited inclusion in this primary research paper and would be of great interest to PLOS One readers.

Q7: Why was glucose uptake represented in units of volume? How can this make sense? It seems to assume that cells import glucose by pinocytosis of the buffer or medium surrounding them.

Q7: RESPONSE: 

We have now more clearly explained the expression of results section:

Line 339- 

The radioactivity (dpm) accumulated in the cells per mg of protein was expressed as a function of the radioactivity in a �l of the initial accumulation buffer to calculate the volume of distribution (Vd; ml/mg of cellular protein) of the [14C]D-glucose or [3H]mannitol as shown in the equation below.

Vd=(dpm per mg ofprotein)/█(dpm per µl of accumulation buffer)

The Vd for [14C]D-glucose can be corrected with the Vd for [3H]mannitol (marker for non-specific binding) at each time point. Values were expressed as a mean±SEM. The [3H]mannitol values can also be used to confirm membrane integrity of each monolayer. [3H]mannitol is a well-established marker of BBB integrity (Kadry et al., 2020). This method allows us to compare the accumulation of a radiolabelled test molecule into human brain endothelial cells after a set of different experimental conditions such as self-inhibition (ie. plus unlabelled glucose) or cross-inhibition (i.e. plus unlabelled amisulpride) studies. In this way we can explore the means of cell entry and even cell exit of the test molecules. Movement across the cell membranes may be by passive diffusion, transporter mediated or indeed involve vesicles which may be receptor mediated.

Q8 Suppl. Fig. 4 is troubling. Many of the 5xFAD mice appear to lack APP, as does the “positive control” 3xTg. Some of the samples are even inconsistent between the two gels: M4 and M5 have no APP in the first membrane but robust levels in the second, and M3 is the opposite. It is very difficult to understand how the authors can claim “WB analysis validated the 5xFAD mouse model for our study.” [Lines 1126–1127]

Q8 RESPONSE:

Thank you for noticing this.

We have replaced the supplementary Figure 4 images with higher resolution images. In these images you can see the APP in the 3xTG lane. It is noted it is faint.

We have also renumbered the lanes and the mice. There were two cohorts of mice and the mice were given the same identifier in the initial Supplementary Figure 4, hence the confusion. We have corrected this. 

Q9: It is inappropriate to heat samples to 95 deg-C before western-blot analysis of membrane transport enzymes. Such high heat causes stable aggregation of GLUT1 and P-gp, retarding entry of a large fraction of these proteins into polyacrylamide gels. Suppl. Figs. 6 and 12 makes it clear that GLUT1 was not successfully detected or quantified in this study, rendering the “data” in Fig. 10 and Suppl. Fig. 9 meaningless.

Q9 RESPONSE: 

The antibody manufacturer advises against boiling the samples (https://www.abcam.com/products/primary-antibodies/glucose-transporter-glut1-antibody-epr3915-ab115730.html#lb). Yet due to sensitivity issues with a previous GLUT1 antibody used, we had tested in the lab, heating (95°C or 100°C), and not heating (room temperature), and the heating condition was picked due to the stronger signal obtained. The heat-related issue has now been noted in the methods (Lines 268-274) and discussion (Line 1256).

Lines 268-274:

‘Optimization assays were performed to compare samples heated at 95°C for 5 minutes to samples left at room temperature. The heated samples gave a stronger signal. Heating samples has previously been utilized for the Recombinant Anti-Glucose Transporter GLUT1 antibody and an anti-Pgp antibody (Ji et al., 2023; Morimoto et al., 2022). 

It has also been noted in the discussion.

Line 1256: This could be related to the effects of the thermal processing of the samples, which could have prevented some of the protein from entering the gel and/or production of multimers which can be mistaken for background. 

Q10: Suppl. Table 2 indicates 9 and 14 samples were available for controls and AD, respectively. But not all of these were analyzed for every protein target. How did the authors select those which were omitted? Was there randomization?

Q10 RESPONSE: 

This was due to the availability of certain processed samples in different time frames. They were not deliberately omitted it was just availability of processed samples.

We have now provided more details of the specific cases used for each protein in the Supplementary Table 7, S1 File.

We have mentioned this on Line 981-983: Details of the specific samples used in the Western blot studies are tabulated in Supplementary Table 7, S1 File.

Q11: A statistician should be consulted regarding the use of t tests for analyses that include several brain regions or other variables that result in multiple measurements (e.g., Figs. 7 and 10). Most statisticians would caution against this due to the introduction of risk for Type-1 error.

Q11 RESPONSE:

Interpretations/conclusions

FIGURE 7

-Fig 7A: Normal two-way ANOVA or repeated measures requires the same n of data points for each group, so when some data points are missing, GraphPad Prism v9 statistical analysis software performs a mixed-effects analysis. In the case of the mouse perfusion data, the data was arranged for repeated measures analysis (since we have multiple samples from the same mouse) but was analysed with mixed-effects analysis due to the missing values.

-We have explained more clearly the types of statistical tests utilized in the Figure 7.

We now state in the text:

-Lines 522-529: The difference between WT and 5xFAD mice in the [3H]amisulpride uptake in the different brain regions was analysed as repeated measures data by fitting a mixed-effects model with Holm-Sidak post hoc test. The same tests were used to analyse the difference in the [14C]sucrose brain uptake between WT and 5xFAD mice. Statistical analysis of the [3H]amisulpride uptake (corrected for [14C]sucrose) into the capillary depletion analysis samples, which included the homogenate, the supernatant or the endothelial cell enriched pellet samples from WT and 5xFAD mice were analysed by unpaired Student’s t-test.

-We now state in Figure 7 legend:

Lines 908-915: ‘A: The effect of genotype in the different brain regions was analysed by Mixed-effects analysis with Holm-Sidak post hoc test. B: The effect of genotype on the capillary depletion samples was analysed by unpaired two-tailed Student’s t-test using GraphPad Prism 9. No significant effect was observed except between the supernatant samples of 5xFAD mice compared to WT mice (*P<0.05)’

-We have also added a sentence in the discussion:

Line 1198-1200: However, we could not detect any changes in [3H]amisulpride (corrected for [14C]sucrose) uptake when we compared specific brain region samples in the 5xFAD and WT mice.

FIGURE 10

- Due to individual variation observed in human tissue samples, we chose to analyse the different transporter samples separately using unpaired Student’s t-test to explore the effect of disease on the expression of GLUT1, P-gp or TfR1 expression.

-In the Figure 10 legend (line 965) we have now clarified that the different transporters were not necessarily assessed in the same brain sample. 

Line 981: We now state: Details of the specific samples used in the Western blot studies are tabulated in Supplementary Table 7, S1 File.

Q12“This suggests that antipsychotics exacerbate the cerebral hypometabolism in AD.” This is not a logical inference from the data. An impact of amisulpride on glucose uptake was not demonstrated.

Q12 RESPONSE: We have ‘softened’ this statement in the abstract so that it now states: Line 53:

‘This suggests that antipsychotics could potentially exacerbate the cerebral hypometabolism in AD.’

Q13: Despite the fact that a literature review was a highlighted part of this report, an enormous body of literature regarding glucose transporters and P-gp in AD and mouse models of amyloid accumulation was ignored in the Discussion.

Q13 RESPONSE: 

This manuscript cites 88 references. 

The literature review method described in the manuscript was to identify groups of molecules that interacted with GLUT. It was not used to generate an exhaustive list of papers where glucose transporters and P-gp were examined in human AD and mouse models of amyloid accumulation. 

However, 14 papers which describe glucose transporters and P-gp in AD are described in the discussion. 

For example, references: 77, 78 and 16 focus on the 5xFAD and 3xTG mice models and wild-type controls. 

For example, references 80, 81, 82, 83, 84, 19, 18, 85, 86, 87 and 88 focus on human tissue in AD and age matched controls.

Q14 In the authors’ prior work (PMID: 31842924), amisulpride was ruled out as a P-gp substrate. Thus, it is odd that P-gp was a molecule of some intensive focus here.

Q14 RESPONSE: We continue to consider that amisulpride is not a Pgp substrate. However, we were exploring the possibility that transporter expression is altered in AD and so we wanted to provide evidence that this was possible. As mentioned previously data from human tissue is rare.

Q15 Language usage

Much of the language usage is unsophisticated or irregular.

Lines 44–45: “There was no significant effect of AD on mouse GLUT1 and P-gp BBB expression….” Of course, there is not; mice do not get AD. It well-recognized in the AD field that claiming (or implying) that transgenic models of amyloid accumulation are models of AD is a serious misrepresentation.

Q15 RESPONSE: We have changed this sentence: Lines 44–45: ‘There was no difference in GLUT1 or P-gp BBB expression between WT and 5XFAD mice. In contrast, caudate P-glycoprotein, but not GLUT1, expression was decreased in human AD capillaries versus controls’.

Q16 There is a strange insertion of commas in many places that are completely uncalled for. Some of these convey an incorrect meaning to the sentences in which they appear. For instance, a comma should only separate a phrase from the item it follows if it is a nonessential phrase, i.e. one that applies to all such items. Thus, in Lines 83–84, “Amisulpride is a second-generation antipsychotic – a substituted benzamide derivative, which is a highly selective dopamine D2 receptor antagonist,” the comma indicates that all substituted benzamide derivatives are highly selective D2R antagonists. That, of course, is not true. The authors are attempting to convey that amisulpride is special among substituted benzamides in its ability to antagonize D2Rs, and that requires the absence of a comma after “derivative.”

Q16 RESPONSE: We have removed the comma after derivative (Line 87).

Q17 The complementary error is made in the sentence that appears in Lines 94–97. Here, comma is required after “(BBB).” Otherwise, the meaning conveyed is that there is a special subset of the BBB which controls the entry of drugs into the brain, whereas the truth of the matter is that the entire BBB contributes to this restriction. Likewise, the omission of a comma in Lines 111–112 leaves the impression that there is a class of glucose transporters that serve to induce hyperglycemia.

Q17 RESPONSE: We have added a comma on line 101 and on line 127.

Q18 In addition, use of the “Oxford comma” is inconsistent.

Semicolons are also used unnecessarily, as in Line 88, where a comma would be appropriate.

Q18 RESPONSE: We have replaced the semicolon with a comma. Line 92

Q19 In Lines 100–102, an incomplete sentence is inserted.

Q19 RESPONSE: Thank you. We have now updated and completed the sentence (Line 106):

 ‘For example, differences in the expression of the multi-drug and toxin extrusion protein 1 (MATE1; SLC47A1)) protein and the plasma membrane monoamine transporter (PMAT; SLC29A4) protein in brain capillaries isolated from healthy individuals compared to AD patients have been observed (16).

Q20 Line 211: “plastic wear” should be “plasticware.”

Q20 RESPONSE: Thank you. We have corrected this. (now Line 235).

Q21 What does the “self” in “self-inhibition studies” mean? Glucose? If it is unlabeled glucose, it is not “self” with respect to tritiated glucose. This is strange terminology for a cold-competition experiment.

Q21 RESPONSE:

This terminology allows separation of two types of cold competition experiments so self-inhibition experiments (i.e. radiolabelled glucose and unlabelled glucose) from cross-inhibition experiments (i.e. radiolabelled glucose and unlabelled substrates and inhibitors). Self-inhibition experiments allow determination of transporter saturability and the measurement of kinetic constants (Km and Vmax) if appropriate. Cross-competition experiments allow identification of the transporter, determination of transporter sensitivity to drugs, determination of transporter ion dependence and determination of transporter inhibition constants (Ki) as required.

We have now added a sentence to explain this in more detail. (Line 351).

Q22 Lines 276–277: The following is odd: “The 4 mM total concentration of non-labelled D-glucose included the standard 1.1mM non-labelled D-glucose normally present in accumulation buffer.” Are the authors attempting to convey that they did not add an additional 4 mM glucose on top of glucose that pre-existed in the buffer at 1.1 mM? Why would anyone assume that they did? The meaning of “total” should be clear to any reader.

Q22 RESPONSE: We think it is open to misinterpretation, so we prefer to include this sentence to avoid doubt. We have, however, refined the explanation on line 320. See response to Q23.

Q23 The sentence that follows that one is also odd: “Radioactivity uptake was compared to the control condition in which the cells were treated with accumulation buffer of the same composition apart from the non-labelled D-glucose content which was 1.1 mM (the standard non-labelled D-glucose concentration in the accumulation buffer).” Is this to say that the control was exactly the same as the experimental treatments except that the concentration of non-labelled glucose was lower by 1.1 mM (i.e., totaled 2.9 mM)? If they are attempting to convey that the control condition was identical except that the concentration of non-labelled glucose was 1.1 mM (i.e., lowered by 2.9 mM), a comma would be needed after “content.” Regardless, it is not clear how this is a control. Is it meant to be a positive control under the assumption that more radiolabeled glucose would be taken up when the unlabeled competitor was at a lower concentration? This is a very strange way to describe a cold-competition experiment. Moreover, a 3.6-fold elevation of cold competitor is hardly enough to achieve a blockade of uptake.

Q23 RESPONSE: We have now modified this sentence. 

It is now as follows: Line 320:

‘Radioactivity uptake was compared to the control condition in which the cells were treated with accumulation buffer of the same composition apart from the additional 2.9mM non-labelled D-glucose of the test condition’.

It is a baseline control. We would like to highlight that we have referred to preliminary experiments where we explored the use of higher concentrations of glucose i.e. 4, 6 and 9 mM in the method (lines 330-332) we state: 

‘The non-labelled D-glucose concentration for the treatment condition (4 mM) was selected based on previous reports that normal plasma glucose concentrations are kept in a narrow range between 4 and 8 mM in healthy subjects (41) and on preliminary experiments studying the effect of 4, 6, and 9 mM non-labelled D-glucose on the membrane integrity of hCMEC/D3 cells (see Supplementary Fig 2, S1 File)’.

In the supplementary file on page 18 we explain:

‘There was no statistically significant effect of non-labelled glucose at a total concentration of 4 mM, 6 mM or 9 mM glucose on cell membrane integrity as measured by [3H]mannitol. However, individual wells in the 9 mM non-labelled glucose had very high Vds, suggesting decreased cell membrane integrity in those wells. As a result, 4 mM non-labelled glucose was selected to test saturable transport of glucose in subsequent accumulation assays’.

We have also taken this opportunity to remove one of the two Supplementary figures in figure 2 as it is preliminary data and the complete data set is illustrated in Figure 5A of the main manuscript. We have updated the text in the supplementary file accordingly.

Q24 The formatting of the manuscript was very unusual. While the figures were compiled at the end, figure legends were interspersed throughout the main body of the text. This made the paper rather difficult to read/evaluate.

Q24 RESPONSE: This type of formatting is requested by the journal.

Q25 Reviewer #3: The manuscript titled "Interaction of amisulpride with GLUT1 at the blood-brain barrier. Relevance to

Alzheimer’s disease" by Boyanova and colleagues investigated the possible interactions of amisulpride (an antipsychotic). Overall, it is a very good manuscript but has some short-term. GLUT1 and GAPDH on the same blot at such a low resolution, but also the absence of GLUT1 band in HEK293 cells. HEK293 cells express a non-negligible amount of GLUT1, which suspects that the antibody is not suited or not good enough. I recommend to perform this blot with SA0377, a Rb monoclonal GLUT1 antibody targeting the C-terminal fraction of the transporter.

Q25 RESPONSE: The GLUT1 Western blots are found in Supplementary Figure 6 and Supplementary Figure 12. In supplementary figure 6 tubulin was used as a loading control. In supplementary figure 12 GAPDH was used as a loading control. Importantly HEK 293 was as specified in the legend used as a negative control. We agree, as the reviewer states, it does not express GLUT1. We did trial a different GLUT1 antibody, but this required a high amount of protein per well. So we selected #ab115730 for further studies. We have now updated Supplementary Fig 12 with higher resolution blots. We have also taken the opportunity to add the tubulin antibody information to the Supplementary Table 1.

Q26 Minor issue: Please show the data as Mean+SD, also provide the P-value if P<0.10 so the reader can assess is there is a trend or not to significance, and if the authors have the Km (substrate) and IC50 (inhibitor) for GLUT1 of the compounds listed. These can help the reader appreciated the selectivity and affinity of these compounds to GLUT1.

Q26 RESPONSE: 

26a) We have presented the data as dot plots and bar graphs reporting the mean ±SEM (Fig 2, Fig 4, Fig 5, Fig 7, Fig 10, Fig S2, Fig S3, Fig S9, Fig S10). The dot plots allow us to observe the variability of the data to the mean (similar to the standard deviation) and the standard error allows us to measure how far the sample mean is likely to be from the true population mean. We have also reported median values where appropriate (Table 1). So as we report the data as dot plots as well as bar graphs we prefer to present the data as mean ±SEM

26b) We had included the IC50 for the antipsychotics that interact with GLUT (Table S6). However, we now specifically mention this in the results text Line 827:

‘Literature review confirmed that other anti-psychotics inhibit glucose uptake at these concentrations. The half-maximal inhibitory concentration (IC50) of these glucose-transport inhibiting antipsychotics ranging from 2-100 �M (Supplementary Table 6, S1 File).’

We now also specifically mention this in the discussion text Line 1130:

‘Although it is noted that other anti-psychotics drugs have been shown to inhibit glucose uptake at these test concentrations (2-100�M) (Supplementary Table 6, S1 File)’.

26c) We have reported different levels of signicance throughout the manuscript. However, we had only placed this information in the text. We have now repeated this information in the figure legends where appropriate e.g. Fig 2 legend, Fig 4C legend, Supplementary Figure 3.

---

## [Decision Letter · Decision Letter 1]

10 Aug 2023

PONE-D-23-12121R1

Interaction of amisulpride with GLUT1 at the blood-brain barrier. Relevance to Alzheimer’s disease

PLOS ONE

Dear Dr. Thomas,

Thank you for submitting your manuscript to PLOS ONE. After careful consideration, we feel that it has merit but does not fully meet PLOS ONE’s publication criteria as it currently stands. Therefore, we invite you to submit a revised version of the manuscript that addresses the points raised during the review process.

We look forward to receiving your revised manuscript.

Kind regards,

Mária A. Deli, M.D., Ph.D.

Academic Editor

PLOS ONE

Journal Requirements:

Additional Editor Comments:

Two reviewers have accepted the manuscript and revision 1, respectively. Reviewer 3 made further comments on revision 1. The selection of the methods and models to study glucose uptake in culture and in mice is scientifically justified. I suggest the authors to cite two further papers in which in situ brain perfusion is used and brain glucose uptake is given as brain distribution volume (Pan et al. 2022 PMID: 36559296; Shah et al. 2015 PMID: 25925411). Please correct the incomplete sentence and remove the duplicate reference.

Reviewers' comments:

Reviewer's Responses to Questions

**Comments to the Author**

1. If the authors have adequately addressed your comments raised in a previous round of review and you feel that this manuscript is now acceptable for publication, you may indicate that here to bypass the “Comments to the Author” section, enter your conflict of interest statement in the “Confidential to Editor” section, and submit your "Accept" recommendation.

Reviewer #2: (No Response)

Reviewer #3: All comments have been addressed

2. Is the manuscript technically sound, and do the data support the conclusions?

Reviewer #2: No

Reviewer #3: Yes

3. Has the statistical analysis been performed appropriately and rigorously? 

Reviewer #2: Yes

Reviewer #3: Yes

4. Have the authors made all data underlying the findings in their manuscript fully available?

Reviewer #2: No

Reviewer #3: Yes

5. Is the manuscript presented in an intelligible fashion and written in standard English?

Reviewer #2: Yes

Reviewer #3: Yes

6. Review Comments to the Author

Reviewer #2: This manuscript has been revised by making minor changes to the text and some of the supplemental figures.

The most significant problems with the manuscript are:

1. The unusual and unverified “in situ perfusion” assay of brain glucose uptake. This is an indefensible assay for CNS glucose uptake and utilization.

2. The inability of the authors to detect and quantify GLUT1 by western blot.

3. The use of the hCMEC/D3 cell line to model glucose uptake by cerebral endothelial cells. Transformed cells have dramatically different glucose utilization, and this likely explains why assays with this line did not support the hypothesis inspired by in silico approaches and prior evidence.

Together, these issues disqualify the study.

The author’s wrote an extensive response to the reviewers. Some of the authors’ responses are provided below in quotation marks, followed by this reviewer’s responses introduced by asterisks:

“Firstly, we are not saying that amisulpride may alter glucose uptake by GLUT1 and that this applies to other antipsychotic drugs.”

* From the Discussion: “Our in silico molecular docking studies revealed that amisulpride interacts with GLUT1 and could potentially affect glucose delivery to the CNS. …The implications of our findings extend to other antipsychotic drugs.”

“Secondly, we also did not assess ‘enzymes’ we assessed the expression of ‘transporters and receptors’.”

* Transporters (and many receptors) are indeed enzymes. If all 14 authors fail understand this, it is a shocking and troubling commentary on biochemical literacy among researchers.

“Thirdly, the physicochemical characterization data does not allow us to rule out that

amisulpride interacts with GLUT.”

* Of course it doesn’t. But scientific publishing consists of reporting what *has* been supported by evidence, not speculation about what has not. If we were to publish everything that has not been disproved, we would quickly consume the bandwidth of the entire internet.

“However, the plasma used in the perfusion assay is artificial and as a result the concentration of glucose in the plasma is stable throughout the experiment.”

* The problem with anesthetics is not with the effects of somatic physiology on blood glucose levels; the problem is that these drugs alter the function of glucose transporters and other mechanisms directly involved in glucose utilization in the CNS.

“On line 450 we state that the: ‘The mouse brain was in situ perfused via the heart with artificial plasma.’”

* That line appears under the subsection titled “Western blot studies.” (Moreover, “heart” is rather vague. Perfusion via the left ventricle would be quite different from perfusion via other chambers.) The subsection titled “In situ brain perfusions” made no mention of the perfusion route in the original document, so it seems that this was a fair criticism.

“We would like to highlight the following paragraph to the reviewers [regarding the suitability of the hCMEC/D3 cell line for glucose uptake assays]:

* The authors provide some information about the various glucose transporters expressed by this cell line and conclude “there is consensus that GLUT1 is the most highly expressed transporter in the hCMEC/D3 cells and the BBB.” The existence of GLUT1 in hCMEC/D3 cells was never disputed. But this alone does not make the cell line sufficiently similar to cerebral endothelial cells. The poor relevance of hCMEC/D3 cells may explain the fact that the authors “could not detect any interaction of amisulpride with GLUT1 in this assay.” But the authors would rather defend their poor experimental design than develop a different one that would defensibly test their in silico predictions. Moreover, the authors failed to address this reviewer’s chief concern, namely that oncogenic transformation with viral large T antigen dramatically alters glucose metabolism. The authors might be advised to research information regarding the Warburg shift.

“We have now more clearly explained the expression of results section [regarding the report of glucose uptake in units of volume]. …In this way we can explore the means of cell entry and even cell exit of the test molecules. Movement across the cell membranes may be by passive diffusion, transporter mediated or indeed involve vesicles which may be receptor mediated.”

* The authors go to great pains to explain a frankly inane computation that relates the amount of radiolabeled glucose found in the cellular compartment to the volume that it originally would have occupied whilst diluted at its original concentration in the assay medium. It seems that the authors need to have explained for them the theory that transporters (e.g., GLUT1) specifically arrest one single molecule of glucose at a time and move it, independent of the aqueous solution in which it is dissolved, from extracellular fluid to cytosol. This has no relationship to volume whatsoever. To report the results in terms of volume suggests that the cell is drinking a liquid that contains a constant concentration of glucose. Indeed, the authors suggest this with their mention of vesicular transport. In 39 years of intense study of cell biology, this reviewer has never encountered evidence that cells emit vesicles to scoop up their liquid surroundings and then transport those vesicles back into their interior; this would be an entirely novel means of import. Regardless, it would also be quite irrelevant to the function of GLUT1, which is what these experiments are intended to address.

“We have replaced the supplementary Figure 4 images with higher resolution images. In these images you can see the APP in the 3xTG lane.”

* Image resolution was not the problem. The problem was that there was an apparent discrepancy between nominal genotype and the appearance of APP. The authors reply that the issue was related to sloppiness, not something one likes to hear from a scientist.

“The antibody manufacturer advises against boiling the samples…. Yet due to sensitivity issues with a previous GLUT1 antibody used, we had tested in the lab, heating (95°C or 100°C), and not heating (room temperature), and the heating condition was picked due to the stronger signal obtained.”

* The fact remains that the blots in Suppl. Fig. 6 show that GLUT1 was not effectively detected by these methods and therefore could not have been accurately quantified. This was also noted by Reviewer 3. It is impossible to surmise what “band” the authors may have tried to quantify for Fig. 10. The authors also misunderstood the comments from Reviewer 3 on this topic. S/he stated that GLUT1 should have been detected in HEK293 cells. Inexplicably, the authors reply, “HEK 293 was as specified in the legend used as a negative control. We agree, as the reviewer states, it does not express GLUT1.” This is refuted not only by Reviewer 3 but by Zambrano et al. (2010, PMID: 20506349), Castro et al. (2008, PMID: 18506475), and Kraft et al. (2015, PMID: 26248369), to name a few.

“This [difference in sample sizes across different outcome measures] was due to the availability of certain processed samples in different time frames. They were not deliberately omitted it was just availability of processed samples.”

* The authors have failed to address whether there may have been biases in the conditions that caused these differences in “availability of processed samples.” The reviewer asked whether any randomization was used; evidently, there was none.

* “Self-inhibition” has been retained in reference to the cold-competition assay. This reviewer still feels this is non-standard (indeed, unprecedented) terminology.

* The Oxford comma is still used inconsistently. This may seem like a trivial matter, but the inconsistency often requires the reader to read a sentence twice to understand the meaning.

* There is still at least one incomplete sentence (Lines 1099–1101).

* References 23 and 27 duplicate one another.

Reviewer #3: (No Response)

7. PLOS authors have the option to publish the peer review history of their article (what does this mean?). If published, this will include your full peer review and any attached files.

Reviewer #2: No

Reviewer #3: **Yes: **Abraham Al-Ahmad (First and Last Name)

---

## [Author Response · Author response to Decision Letter 1]

13 Sep 2023

We have uploaded a word document listing our responses to comments. This uploaded document also includes a Table and a Figure. This information can not be displayed in this box.

We would recommend you refer to the uploaded document.

RESPONSES TO REVIEWERS COMMENTS:

COMMENTS FROM EDITORIAL TEAM:

Additional Editor Comments:

Two reviewers have accepted the manuscript and revision 1, respectively. Reviewer 3 made further comments on revision 1. The selection of the methods and models to study glucose uptake in culture and in mice is scientifically justified. I suggest the authors to cite two further papers in which in situ brain perfusion is used and brain glucose uptake is given as brain distribution volume (Pan et al. 2022 PMID: 36559296; Shah et al. 2015 PMID: 25925411). Please correct the incomplete sentence and remove the duplicate reference.

RESPONSE FROM AUTHORS:

1. We would firstly like to thank the editor and editorial team for their time spent reviewing the manuscript.

2. We would like to mention that we measured amisulpride and sucrose, not glucose, uptake using the in situ brain perfusion method. 

3. We have now added two papers to justify the in situ brain perfusion method. These are papers which use the same method, are from our group at King’s College London and one also examines the test molecule, amisulpride, but in a different animal model of AD. The papers are:

a) Sekhar, G.N., Fleckney, A.L., Boyanova, S.T., Rupawala, H., Lo, R., Wang, H., Farag, DB, Rahman, KM, Broadstock, M., Reeves S. and Thomas, S.A. Region-specific blood–brain barrier transporter changes leads to increased sensitivity to amisulpride in Alzheimer’s disease. Fluids Barriers CNS 16, 38 (2019). https://doi.org/10.1186/s12987-019-0158-1.

b) Sanderson, L., Khan A. and Thomas, S.A. Distribution of suramin, an antitrypanosmal drug, across the blood-brain and blood-CSF interfaces in wild-type and P-glycoprotein transporter-deficient mice. Antimicrobial Agents and Chemotherapy 51: 3136-3146 (2007). https://doi.org/10.1128/aac.00372-07

4. We have corrected the sentence and removed the duplicate reference.

5. We have also further proofread the manuscript and added missing abbreviations (TBS-T, GAPDH), corrected and moved an ethics statement (page 10, line 233), clarified further some methological details (page 24, line 619-622). Specified that the antipsychotics were GLUT interacting where appropriate (so not simply stated antipsychotics e.g. Table 1). We have also correctly placed the endothelial cell label in Fig 8a.

Reviewer #2: This manuscript has been revised by making minor changes to the text and some of the supplemental figures. The most significant problems with the manuscript are:

1. The unusual and unverified “in situ perfusion” assay of brain glucose uptake. This is an indefensible assay for CNS glucose uptake and utilization.

RESPONSE FROM AUTHORS: 

We would firstly like to thank the reviewer for their time spent reviewing the manuscript.

We did not measure glucose uptake using the in situ brain perfusion method in this study.

We assessed amisulpride and sucrose uptake.

We have now added two method references to the paper (as described above). These papers justify and verify the use of the method used in the submitted manuscript. As mentioned by the editor this method has also previously been used by other research groups (Pan et al. 2022 PMID: 36559296; Shah et al. 2015 PMID: 25925411).

2. Reviewer #2: The inability of the authors to detect and quantify GLUT1 by western blot.

RESPONSE FROM AUTHORS: 

We have been able to detect GLUT1 protein expression. We comment on this in more detail in response to question 14 from reviewer #2.

3. Reviewer #2 The use of the hCMEC/D3 cell line to model glucose uptake by cerebral endothelial cells. Transformed cells have dramatically different glucose utilization, and this likely explains why assays with this line did not support the hypothesis inspired by in silico approaches and prior evidence.

RESPONSE FROM AUTHORS:

Although we could not find a published paper that examined glucose utilization in the hCMEC/D3 cells we have now further explained the limits of the assay in the paper. We have mentioned:

Page 48, line 1241: 

‘However, these results are difficult to interpret conclusively as the immortalized cell line may not utilize or transport glucose in the same manner as cells in vivo’. 

4. Reviewer #2 Together, these issues disqualify the study.

RESPONSE FROM AUTHORS:

We disagree that the study should be disqualified.

Reviewer #2 is incorrect, please see responses above and those from the editor and the other reviewers.

5. Reviewer #2 The author’s wrote an extensive response to the reviewers. Some of the authors’ responses are provided below in quotation marks, followed by this reviewer’s responses introduced by asterisks:

“Firstly, we are not saying that amisulpride may alter glucose uptake by GLUT1 and that this applies to other antipsychotic drugs.”

* From the Discussion: “Our in silico molecular docking studies revealed that amisulpride interacts with GLUT1 and could potentially affect glucose delivery to the CNS. …The implications of our findings extend to other antipsychotic drugs.”

RESPONSE FROM AUTHORS:

The statements above have been taken out of context and the original question and the full explanations provided in the first response to reviewers comments are not being considered in their entirety.

The original query was:

Q3 revision one comments . Reviewer #2: This manuscript reports a gravely flawed analysis of the interactions of an antipsychotic drug (amisulpride) with aspects of cerebrovascular transport, including attempts at relevance to Alzheimer’s disease. The approach included literature review to identify and characterize confirmed and proposed antipsychotics that interact with glucose transporter 1 (GLUT1), in-silico modeling of amisulpride with GLUT1, in-vitro tests of glucose uptake in a transformed endothelial cell line, and a in-vivo perfusion assay of amisulpride uptake into the brain of a mouse model of amyloid accumulation. Levels of some transport enzymes (including GLUT1) in mouse and human brain were also assayed, and the ultrastructural integrity of cerebrovascular structure was evaluated in an Alzheimer’s case. Despite the lack of physicochemical data to this effect, the authors conclude that amisulpride may alter glucose uptake by GLUT1 and that this necessarily applies to other antipsychotic drugs. They also conclude that levels of GLUT1 are not altered in AD or a mouse model of amyloid accumulation and that P-gp levels are lower in AD brains but only in the caudate.

Our full response to this was as follows:

Firstly, we are not saying that amisulpride may alter glucose uptake by GLUT1 and that this applies to other antipsychotic drugs.

We have in silico evidence that amisulpride interacts with GLUT1 and other groups have evidence that other antipsychotic drugs interact with glucose transporters including GLUT1. In fact, 11 antipsychotics were identified which interact with glucose transporters (see Supplementary Table 5, S1 File).

So there is evidence from other sources that would suggest amisulpride has potential to interact with GLUT. 

Secondly, we also did not assess ‘enzymes’ we assessed the expression of ‘transporters and receptors’.

Thirdly, the physicochemical characterization data does not allow us to rule out that amisulpride interacts with GLUT.

Regarding the physicochemical characteristics:

-The +1 charge of amisulpride is similar to the other antipsychotics that interact with the glucose transporters. 

-In addition, we identified one GLUT1 substrate that has a +1 charge so there is evidence that other types of molecules that interact with GLUT 1 have a similar charge.

-The MW of amisulpride suggested it would more likely be an inhibitor of GLUT1 transport than a GLUT1 substrate due to its similarity to other GLUT1 inhibitors.

6. Reviewer #2

“Secondly, we also did not assess ‘enzymes’ we assessed the expression of ‘transporters and receptors’.”

* Transporters (and many receptors) are indeed enzymes. If all 14 authors fail understand this, it is a shocking and troubling commentary on biochemical literacy among researchers.

RESPONSE FROM AUTHORS: 

Transporters, receptors and enzymes have different gene names. Transporters are not enzymes.

Transporters carry substrates across a membrane.

Enzymes catalyse reactions and will alter the structure of a substrate.

7. Reviewer #2

“Thirdly, the physicochemical characterization data does not allow us to rule out that

amisulpride interacts with GLUT.”

* Of course it doesn’t. But scientific publishing consists of reporting what *has* been supported by evidence, not speculation about what has not. If we were to publish everything that has not been disproved, we would quickly consume the bandwidth of the entire internet.

RESPONSE FROM AUTHORS: 

We have described in silico evidence, physicochemical evidence, and literature review evidence, which suggests that amisulpride can interact with GLUT1.

8. Reviewer #2

“However, the plasma used in the perfusion assay is artificial and as a result the concentration of glucose in the plasma is stable throughout the experiment.”

* The problem with anesthetics is not with the effects of somatic physiology on blood glucose levels; the problem is that these drugs alter the function of glucose transporters and other mechanisms directly involved in glucose utilization in the CNS.

RESPONSE FROM AUTHORS: 

As mentioned earlier in response to question 1 (reviewer#2) we do not examine glucose transport or glucose utilization using the in situ brain perfusion method. 

However, both the control and test results were from animals that were anaesthetized using the same anaesthetic /route of administration so the comparisons and conclusions are appropriate.

9. Reviewer #2

“On line 450 (now 469) we state that the: ‘The mouse brain was in situ perfused via the heart with artificial plasma.’”

* That line appears under the subsection titled “Western blot studies.” (Moreover, “heart” is rather vague. Perfusion via the left ventricle would be quite different from perfusion via other chambers.) The subsection titled “In situ brain perfusions” made no mention of the perfusion route in the original document, so it seems that this was a fair criticism.

RESPONSE FROM AUTHORS: We did alter the manuscript and we did thank the reviewers for their constructive comments so we have already agreed that this point was valid.

We have now more clearly specified the route of perfusion in the Western blot studies section and we have also now mentioned that the method is described in more detail elsewhere in the document.

Page 19, line 470: Specifically, we state: ‘The mouse brain was in situ perfused via the left ventricle of the heart with artificial plasma containing radiolabelled amisulpride and radiolabelled sucrose for 10 minutes as described below.’

10. Reviewer #2

“We would like to highlight the following paragraph to the reviewers [regarding the suitability of the hCMEC/D3 cell line for glucose uptake assays]:

* The authors provide some information about the various glucose transporters expressed by this cell line and conclude “there is consensus that GLUT1 is the most highly expressed transporter in the hCMEC/D3 cells and the BBB.” The existence of GLUT1 in hCMEC/D3 cells was never disputed. But this alone does not make the cell line sufficiently similar to cerebral endothelial cells. The poor relevance of hCMEC/D3 cells may explain the fact that the authors “could not detect any interaction of amisulpride with GLUT1 in this assay.” But the authors would rather defend their poor experimental design than develop a different one that would defensibly test their in silico predictions. Moreover, the authors failed to address this reviewer’s chief concern, namely that oncogenic transformation with viral large T antigen dramatically alters glucose metabolism. The authors might be advised to research information regarding the Warburg shift.

RESPONSE FROM AUTHORS: Please see response to reviewer #2 question number: 3 (second set of comments). 

11. Reviewer #2

“We have now more clearly explained the expression of results section [regarding the report of glucose uptake in units of volume]. …In this way we can explore the means of cell entry and even cell exit of the test molecules. Movement across the cell membranes may be by passive diffusion, transporter mediated or indeed involve vesicles which may be receptor mediated.”

* The authors go to great pains to explain a frankly inane computation that relates the amount of radiolabeled glucose found in the cellular compartment to the volume that it originally would have occupied whilst diluted at its original concentration in the assay medium. It seems that the authors need to have explained for them the theory that transporters (e.g., GLUT1) specifically arrest one single molecule of glucose at a time and move it, independent of the aqueous solution in which it is dissolved, from extracellular fluid to cytosol. This has no relationship to volume whatsoever. To report the results in terms of volume suggests that the cell is drinking a liquid that contains a constant concentration of glucose. Indeed, the authors suggest this with their mention of vesicular transport. In 39 years of intense study of cell biology, this reviewer has never encountered evidence that cells emit vesicles to scoop up their liquid surroundings and then transport those vesicles back into their interior; this would be an entirely novel means of import. Regardless, it would also be quite irrelevant to the function of GLUT1, which is what these experiments are intended to address.

RESPONSE FROM AUTHORS: 

The accumulation assay results are expressed as a volume of distribution (ml.g-1 of protein). This is an accepted and standardized expression of the results/data.

We do not measure the transport of one molecule by one transporter.

In this assay the concentration of the test molecule is measured and used to calculate a volume of distribution.

We do not specifically state that vesicles are released into the exterior. Although the contents of vesicles can be released into the exterior. 

12. Reviewer #2

“We have replaced the supplementary Figure 4 images with higher resolution images. In these images you can see the APP in the 3xTG lane.”

* Image resolution was not the problem. The problem was that there was an apparent discrepancy between nominal genotype and the appearance of APP. The authors reply that the issue was related to sloppiness, not something one likes to hear from a scientist.

RESPONSE FROM AUTHORS: 

The authors have not used the word ‘sloppiness’. An incorrect figure was uploaded to the supplementary file. This has now been corrected. 

13. Reviewer #2

“The antibody manufacturer advises against boiling the samples…. Yet due to sensitivity issues with a previous GLUT1 antibody used, we had tested in the lab, heating (95°C or 100°C), and not heating (room temperature), and the heating condition was picked due to the stronger signal obtained.”

* The fact remains that the blots in Suppl. Fig. 6 show that GLUT1 was not effectively detected by these methods and therefore could not have been accurately quantified. This was also noted by Reviewer 3. It is impossible to surmise what “band” the authors may have tried to quantify for Fig. 10. 

RESPONSE FROM AUTHORS: 

-Supplementary file 1, Figure 6: We apologise the band label had slipped. This has now been corrected.

-Figure 10 shows: TfR1, GLUT1 and P-gp expression in human frontal cortex and caudate brain capillary lysates from human control and AD age-matched cases. We have now ‘linked’ these graphs to the example immunoblots in the S1 file in the text. The exact figure numbers have now been included in the text page 41-42, lines 1059-1085.

14. Reviewer #2. The authors also misunderstood the comments from Reviewer 3 on this topic. S/he stated that GLUT1 should have been detected in HEK293 cells. Inexplicably, the authors reply, “HEK 293 was as specified in the legend used as a negative control. We agree, as the reviewer states, it does not express GLUT1.” This is refuted not only by Reviewer 3 but by Zambrano et al. (2010, PMID: 20506349), Castro et al. (2008, PMID: 18506475), and Kraft et al. (2015, PMID: 26248369), to name a few.

RESPONSE FROM AUTHORS

Reviewer 3 (first set of comments) stated this: ‘HEK293 cells express a non-negligible amount of GLUT1, which suspects that the antibody is not suited or not good enough. 

Reviewer 2 (second set of comments) then went on to provide three papers where GLUT1 has been detected in HEK 293 cells. These were: Zambrano et al. (2010, PMID: 20506349), Castro et al. (2008, PMID: 18506475), and Kraft et al. (2015, PMID: 26248369). 

a) We firstly apologise for mis-interpreting the antibody sensitivity query from reviewer 3 (first set of comments). The reviewer was, in fact, querying the use of HEK293 cells as a negative control for the expression of GLUT1 in our manuscript. 

b) Reviewer 2 provided papers which suggested that HEK293 cells should, in fact, be a positive control for the expression of GLUT1.

However, two of the three papers that reviewer 2 cited, did not use the Western blot method (Table 1). They measured mRNA by RT-PCR so the reviewer has actually have not provided evidence for the presence of ‘GLUT1 proteins’ in HEK293. It is possible for GLUT1 mRNA to be present, but for GLUT1 protein not to be expressed in cells. So these two papers do not provide evidence for GLUT1 protein expression in HEK293 cells.

The third paper that reviewer 2 cited used the Western blot method, but did not specify the antibody used to detect GLUT1.

Table 1: Publications cited by Reviewer 2.

Publication Protein mRNA HEK293

Kraft et al., (2015

PMID 26248369) Did not describe /utilize the Western blot method. measured mRNA expression levels HEK293 cells were acquired from the American Type Culture Collection

Castro et al. (2008, PMID: 18506475)

 Did not use Western blot. They used RT-PCR so measured mRNA HEK293, ATCC® CRL-1573TM

Zambrano et al. (2010, PMID: 20506349)

 Western blot

Did not mention antibody used. The HEK293 cell line (ATCC1 CRL-1573TM).

c) Another point is the source of the HEK 293 cells. We used commercially prepared human embryonic kidney 293 (HEK293) cell lysates (#ab7902, Abcam, UK) as a negative control for GLUT1. In contrast all three papers in Table 1 used HEK293 cells from ATCC, USA. This may also contribute to differences in GLUT1 expression.

We have now added the source of the HEK293 cells to the manuscript (Page 7, line 150).

d) We originally selected HEK293 lysates as a GLUT1 negative control based on the data reported in Figure 1. So positive and negative controls were chosen on the basis of RNA-seq data from The Human Protein Atlas (https://www.proteinatlas.org/, last accessed: 10/10/2019).

Figure 1: RNA-seq tissue/cell data for GLUT1 reported as mean pTPM (protein-coding transcripts per million), corresponding to mean values of the different individual samples from various tissues or cell lines). GLUT-1 RNA-sequencing results are reported as normalised expression (NX) values. NX value of 1 is defined as the threshold for expression of the corresponding protein (https://www.proteinatlas.org/). Green rectangular showing the positive control we used, red rectangular – showing the negative control.

We have now further justified the use of HEK293 as a negative control for GLUT1 by adding more information about the selection of positive and negative controls in the methods section. In fact, we have now justified our selection of negative and positive controls throughout the manuscript. We have also added appropriate references. These are references 38, 39, 49, 51 and 52.

In hCMEC/D3 cells Page 12 Line 291 now states:

‘Positive and negative controls for transporter expression in hCMEC/D3 cells were chosen based on RNA-seq data from The Human Protein Atlas (38,39). Lysates from human colon adenocarcinoma (Caco-2) cells were used as a positive control, whilst whole cell lysate from human embryonic kidney cell line (HEK-293; (#ab7902, Abcam, UK) were used as a negative control for GLUT1 transporter expression’.

In WT and 5XFAD mice Page 20, line 500-507 now states:

‘Positive and negative controls for transporter expression in the mouse brain capillary lysates were chosen based on RNA-seq data from The Human Protein Atlas (38,49). They included mouse whole kidney lysates as positive controls for PMAT, MATE1 and P-gp expression. HEK293 was also used as a positive control for MATE1 and P-gp. Lysates from cultured human airway epithelial cells (Calu3) was used as a negative control for PMAT (38,50). HL-60 was used as a negative control for MATE1 and P-gp (38,51,52). The positive control for OCT1 was HL-60 and negative control was Calu3. The positive control for GLUT1 expression was Caco2 cell lysates and negative control was HEK293 cell lysates’.

In human brain capillaries Page 26, line 559-563.

‘Positive and negative controls for transporter expression in the human brain capillary lysates were chosen based on RNA-seq data from The Human Protein Atlas (38,49). Caco-2 cell lysates were used as a positive control, and HEK293 whole cell lysate and HL-60 were used as negative controls for GLUT1 transporter expression (38,39). HEK-293 cell lysate was used as a positive control for P-gp transporter expression (38,52).’

e) We have also now mentioned that we also used human promyelocytic leukaemia cell line (HL-60) lysate as a negative control for GLUT1 in our study. Although this is noted in the Supplementary file figure 12 and legend. We had not mentioned this additional negative control in the manuscript text. We have now added the source of these HL-60 cells Page 7, line 139 and mentioned its use as a negative control for GLUT1 expression in the human brain capillary lysates in the manuscript (Page 26, lines 583-588 and Page 42, lines 1081-1083).

15. Reviewer #2

“This [difference in sample sizes across different outcome measures] was due to the availability of certain processed samples in different time frames. They were not deliberately omitted it was just availability of processed samples.”

* The authors have failed to address whether there may have been biases in the conditions that caused these differences in “availability of processed samples.” The reviewer asked whether any randomization was used; evidently, there was none.

RESPONSE FROM AUTHORS: The samples were not randomized. We have now specifically stated this in the manuscript in Fig 10 legend (page 41, line 1067).

16. Reviewer #2

* “Self-inhibition” has been retained in reference to the cold-competition assay. This reviewer still feels this is non-standard (indeed, unprecedented) terminology.

RESPONSE FROM AUTHORS: The authors believe it is appropriate terminology. Self-inhibition terminology is used in the following publications, so it is not unprecedented.

a) Effect of Transport Inhibitors and Additional Anti-HIV Drugs on the Movement of Lamivudine (3TC) across the Guinea Pig Brain Barriers. J. E. Gibbs, T. Rashid and S. A. Thomas

Journal of Pharmacology and Experimental Therapeutics September 1, 2003, 306 (3) 1035-1041; DOI: https://doi.org/10.1124/jpet.103.053827.

b) Dogruel, M., Gibbs, J.E. and Thomas, S.A. (2003), Hydroxyurea transport across the blood–brain and blood–cerebrospinal fluid barriers of the guinea-pig. Journal of Neurochemistry, 87: 76-84. https://doi.org/10.1046/j.1471-4159.2003.01968.x

17. Reviewer #2

* The Oxford comma is still used inconsistently. This may seem like a trivial matter, but the inconsistency often requires the reader to read a sentence twice to understand the meaning.

RESPONSE FROM AUTHORS: 

We have now reviewed each of the commas in the main manuscript and in the supplementary file and we have removed the Oxford comma. 

18. Reviewer #2

* There is still at least one incomplete sentence (Lines 1141-1151).

RESPONSE FROM AUTHORS: 

We have now revised these sentences (now lines 1248-1261).

17. Reviewer #2

* References 23 and 27 duplicate one another.

RESPONSE FROM AUTHORS: 

We have removed the duplication of reference 23.

---

## [Editor Report · Decision Letter 2]

28 Sep 2023

Interaction of amisulpride with GLUT1 at the blood-brain barrier. Relevance to Alzheimer’s disease

PONE-D-23-12121R2

Dear Dr. Thomas,

We’re pleased to inform you that your manuscript has been judged scientifically suitable for publication and will be formally accepted for publication once it meets all outstanding technical requirements.

Kind regards,

Mária A. Deli, M.D., Ph.D.

Academic Editor

PLOS ONE

Additional Editor Comments (optional):

The authors have answered all the remaining questions and further improved the paper. As an expert in the same biomedical field I could judge without further peer review that the revision is complete and I consider the manuscript suitable for acceptance and publishing.
---

## [Editor Report · Acceptance letter]

16 Oct 2023

PONE-D-23-12121R2 

Interaction of amisulpride with GLUT1 at the blood-brain barrier. Relevance to Alzheimer’s disease. 

Dear Dr. Thomas:

I'm pleased to inform you that your manuscript has been deemed suitable for publication in PLOS ONE. Congratulations! Your manuscript is now with our production department. 

Kind regards, 

on behalf of

Prof. Mária A. Deli 

Academic Editor

PLOS ONE